# Towards high resolution, validated and open global wind power assessments

E. U. Peña-Sánchez[1,2,6], P. Dunkel [1,2,6] ✉, C. Winkler [1,2,6], H. Heinrichs [1,3], F. Prinz[1], J. M. Weinand [1], R. Maier [1,2], S. Dickler[1], S. Chen [1,4], K. Gruber [5], T. Klütz [1], J. Linßen [1] & D. Stolten[1,2]

Wind power is widely recognized as a key component of future net-zero energy systems. However, substantial variability among current wind resource and power simulations used for wind farm deployment limits the reliability of siting and system integration decisions. Therefore, we present a transparent, open source, validated and evaluated, global wind power simulation workflow for the renewable energy simulation tool *ETHOS.RESKit*. As the first wind simulation workflow using the new Global Wind Atlas 4 and the fifth iteration of the European Center for Medium-Range Weather Forecasts Reanalysis, it is capable of simulating time-resolved energy output of wind turbines with high spatial resolution and customizable designs for both onshore and offshore wind turbines. The tool has undergone an extensive validation and calibration process using over 18 million global measurements from meteorological masts for wind speed bias correction and 8 million measurements from wind turbine sites across 6 countries during 2002 to 2021. In comparison with measured energy yield from wind turbine sites, we achieve a global average capacity factor mean error of 5.6% and Pearson correlation of 0.844. In addition, we evaluate its performance against annually aggregated energy production data from operational wind farms and country-level wind power generation statistics reported by energy agencies, demonstrating its accuracy across multiple spatial and temporal scales with a mean error of only 0.6%. Additionally, a final calibration step ensures alignment of the simulation with real world statistics. The release of *ETHOS.RESKit* is a step towards a fully open source and open data approach to accurate wind power modeling by incorporating comprehensive simulation advances in one model.

Wind power is placed as one of the largest renewable sources for the upcoming decades[1–4]. Thus, evaluating extractable wind energy resources is essential to develop strategies for the energy systems transformation, for instance in capacity planning, designing adequate market frameworks, or for increasing the speed of planning and permitting[2,4–6]. Being able to accurately assess wind resources ultimately leads to more reliable future energy transformation strategies.

Extractable wind energy resources depend on the location (spatial dependency), on the conditions at a particular time (temporal dependency), and on the wind turbine performance (technology dependency) to translate wind speed's kinetic energy into electricity output. Incorporating these three aspects in one wind energy assessment tool is essential to enhance the robustness and reliability of results. There have been continuous efforts within the renewable energy simulation community to capture these dependencies using

time-resolved and geospatially-constrained wind power simulation models[7].

A methodological gap persists, however, when it comes to the calibration and validation of such wind power and energy simulation models. For example, a previous article[8] found that only 21% of studies assessing large-scale wind resource potentials conduct a validation of the input data. For the open-source simulation tools, the situation is even worse: Whilst *Renewables.ninja* provides a calibration over selected European countries[9] and Murica et al.[10] perform a validation of country-level generation time-series for European countries, no wind energy simulation tool is validated at global scale to the knowledge of the authors. The consequence is a lack in reliability of the simulation results. The present paper addresses this shortcoming by developing a global calibration and validation process which is demonstrated using the example of the open-source *ETHOS.RESKit* but is applicable also to other wind energy simulation tools.

The importance of open-source models and datasets in energy research has gained increased attention, as they facilitate transparency, enable reproducibility of results, and promote collaborative development[8]. Widely used, state-of-the-art, open-source models that account for the three wind power dependencies are *Renewables.ninja*[9], *RESKit* (now *ETHOS.RESKit*)[11], *pyGRETA*[12] and *Atlite*[13] (see Table 1). The first three models use weather data based on MERRA-2 (Modern-Era Retrospective Analysis for Research and Applications v2)[14], additionally *RESKit* and *pyGRETA* take advantage of the higher-resolved Global Wind Atlas (GWA)[15] to increase the spatial resolution to 1 km². *Atlite* employs ERA5 (European Center for Medium-Range Weather Forecasts Reanalysis v5)[16] data, which compared to MERRA-2 has a higher spatial resolution of 0.28° [~ 31 km²] and offers wind speeds at 100 m height instead of 50 m as in the case of MERRA-2.

All four models face two primary limitations: first, the absence or restricted availability of validation procedures and, second, the unaddressed inherited bias from their weather data source as shown by various studies[8,17–23]. The most overlooked aspect is the validation of model outcomes despite its crucial relevance to narrow uncertainties and enhancing the robustness of assessments as emphasized by other authors[7,9]. Validation, as understood by the authors, involves comparing model outcomes with real-world data to assess how accurately the simulation represents actual observed conditions. Only *Renewables.ninja* and the initial *RESKit* model provide a validation procedure at all, but exclusively for European wind production. Moreover, no supplementary validations of these models have been conducted in other regions. Consequently, an evaluation of the models' reliability and performance on a global scale remains an open question. *Renewables.ninja* validated their model results against monthly-aggregated country wind power generation data from the European Network of Transmission System Operators for Electricity (ENTSO-E)[24] as well as nationally aggregated wind power generation data with at least hourly resolution from eight power system operators for eight European countries. The authors[9] found a systematic mean error in wind speeds in MERRA-2 across Europe. Based on this finding, they calibrated the results of their model by incorporating national correction factors. The initial *RESKit* model[11] had been validated against hourly power generation data from two wind parks, resulting in a high Pearson correlation between 0.80-0.88, with total power generation underestimations between 5 and 37%. In addition, the model performance was compared with monthly power generation data from 86 turbines in Denmark, where the majority of deviations in power generation range from −20 to 30%.

The second limitation is defined by the absence or insufficient measures taken to rectify mean errors present in the input data. Previous studies have documented that reanalysis data as well as the GWA inherently contain certain deviations and mean errors. For instance, seasonal and diurnal mean errors in MERRA-2 and ERA5 wind speed

**Table 1 | Comparison of common global open-source wind power models**

| Model | Author(s), year | Data source | Resolution [time, lat/lon] | Turbine modeling characteristics | Validation of results |
|---|---|---|---|---|---|
| *Renewables.ninja* | Staffell & Pfenninger[9], | MERRA-2[14] | 1 hour, 0.5°/0.625° | 141 existing turbine models | Time-resolved country-level aggregated data in eight European countries |
| *RESKit (now ETHOS.RESKit)* | Ryberg et al[11]., 2017 | MERRA-2[14] | 1 hour, 0.5°/0.625° (scaled to 1 km²) | 123 existing turbine models and user-defined configurations on hub height, rotor diameter and capacity to derive synthetic power curves | Two hourly-resolved wind park generation data: one in France and one in the Netherlands and monthly power generation in Denmark |
| *pyGRETA* | Siala & Houmy[12], 2020 | MERRA-2[14] | 1 hour, 0.5°/0.625°(scaled to 1 km²) | Allows user-defined changes in cut-in and cut-out wind speeds and the full-load stage in the power curves | Not provided |
| *Atlite* | Hofmann et al[13]. | ERA5[16] | 1 hour, 0.28125° | 27 existing turbine models | Not provided |

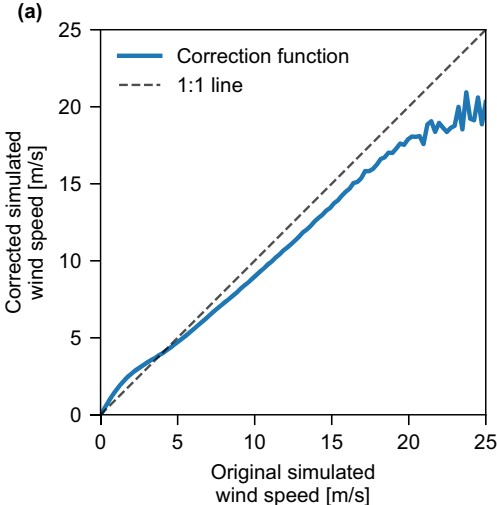

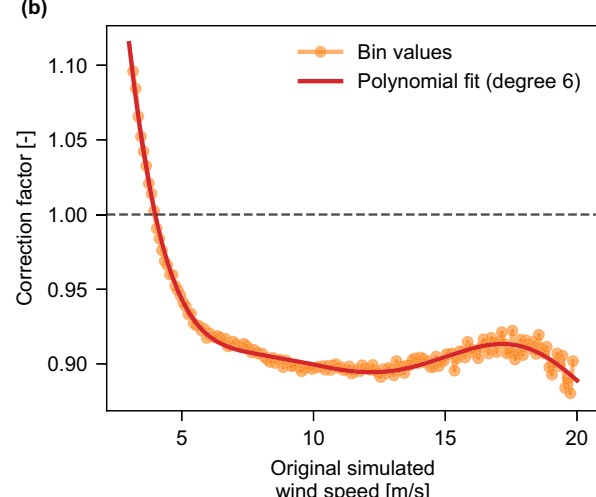

**Fig. 1 | Effect of the value-based wind speed calibration for different simulated wind speeds. a** Corrected versus original simulated wind speeds, with the 1:1 line shown for reference. Wind speeds below approximately 4.0 m/s are adjusted upwards, while higher speeds are adjusted downwards. **b** The correction factor applied as a function of wind speed including a sixth-degree polynomial fit illustrating the magnitude of wind speed correction at different wind speeds. Source data are provided as a Source Data file.

data have been found previously[23], as well as terrain-related deviations when comparing wind speeds from reanalysis data with wind speed measurements[20,22]. Furthermore, statistical comparisons of these two datasets have been conducted[18,21], ultimately concluding that ERA5 exhibited superior performance in comparison to MERRA-2. In addition, global mean errors in wind speeds at 10 m were also found in the GWA4 (see Supplementary Section 1.12). An overall underestimation across the globe is even more pronounced in Eastern Australia, along the US coastlines, in Southern Europe and the Mediterranean, the Siberian Taiga and rain-forested regions, to a lesser extent also throughout Sub-Saharan Africa and South America. Overestimations occur isolated especially in the Northern hemisphere everywhere, but are found in larger quantities throughout continental Asia and in Northern Europe as well as in Canada (see Supplementary Fig. 1). Therefore, the evaluation and subsequent correction of wind speeds derived from reanalysis data and the GWA4 can contribute significantly to the accuracy of wind energy assessments. Notably, although *Renewable.ninja* and the initial *RESKit* model acknowledge such effects and indirectly address them via their validation procedure, none of the listed models has utilized wind speed correction measures to address inherent mean errors in reanalysis data and the GWA.

To cover the existing bandwidth of wind turbine characteristics, simulating as many commercially available wind turbines as possible can support achieving more realistic power generation estimations. As presented in Table 1, most models offer to simulate the performance of such turbines although the available types of turbines vary from 27 to 141. The most flexible approach when it comes to user-defined turbines is provided by the initial *RESKit* model[11] because it is the only model that allows the user to define a synthetic wind power curve, in addition to the ones declared by the manufacturers, based on three wind turbine parameters: hub height, rotor diameter and capacity. This is especially useful for simulating prospective wind turbines, which is often necessary when evaluating future scenarios. In summary, the identified constraints in the reviewed literature comprise the lack of thorough validation encompassing regions beyond Europe, the absence of mean error corrections in the input weather data source, and the lack of incorporating and evaluating the performance of contemporary and prospective wind turbine models.

In this article, we address the above-mentioned limitations of wind power models to enhance their reliability and applicability. Thus,

this study introduces an enhanced expansion to the wind power module of *ETHOS.RESKit*. Our model addresses the identified limitations through extensive validation, global applicability, and the incorporation of more than 800 wind turbine models available to the modeler. To enhance precision, we implement a comprehensive calibration of wind speed data gathered from 213 global weather mast locations in 25 different countries globally, spanning over 8 million hours of observation after filtration, aiming at rectifying systematic wind-speed dependent deviations present in reanalysis data and the GWA4.

Furthermore, we validate the simulated wind power output by comparing it with the actual hourly output from 152 turbines and wind farm sites. Finally, we further validate our model by comparing the outcomes of the simulation of over 30 000 existing global wind farms with over 490,000 turbines with yearly wind power generation estimates derived from statistical analysis, as well as with publicly available country-level hourly wind power generation data. In response to this analysis, we introduce a methodology and provide global capacity factor correction factors as open data to enhance alignment with widely available country-specific wind power generation data.

Therefore, our open-source tool aims to close the outlined gap in literature by first addressing general biases present in reanalysis data and the GWA4 by applying a wind speed dependent wind speed correction and secondly by performing an extensive validation of the model output against various real-world data. Through these rigorous measures, our work contributes significantly to the reliability of future wind power simulations. This contribution is of utmost relevance for the ongoing energy transformation, providing a robust foundation for accurate, open and globally applicable wind energy assessments.

## Results

### Wind speed calibration impact and improvements

The calibration of input wind speeds as described (c.f. Methods Section "Calibration and cross-validation of estimated wind speeds from reanalysis weather data") yielded enhanced performance across all wind dependencies. Figure 1 shows the impact of wind speed calibration by illustrating the changes when applying the value-based wind speed adjustment to the raw simulated wind speeds. Figure 1a shows the result of a calibration applied to simulated wind speeds. The calibration increases wind speeds below approximately 4.0 m/s and decreases those above it. Figure 1b details the non-linear nature of this

**Table 2 | Comparison of key statistical indicators comparing simulated and measured hourly capacity factors from wind farms from two workflow configurations: calibrated and non-calibrated**

| Indicator [unitless] | Calibrated | Non-calibrated | Delta (relative) [%] | Significance for wind energy assessments |
|---|---|---|---|---|
| Measured mean | 0.367 | | - | - |
| Mean | 0.388 | 0.47 | −18.2 | Closer approximation to the total power generation by the turbines allowing for more precise economic estimations such as levelized cost of electricity, return of investment, value of loss load, etc. |
| Mean error (relative) | 0.056 | 0.292 | −80.7 | |
| Perkins skill score | 0.90 | 0.86 | +3.68 | Closer approximation to the power generation stochastic variability allowing for more precise technical considerations design to reduce this type of variability in energy systems such as infrastructure capabilities in storage, transmission, etc. |
| Root-mean square error | 0.196 | 0.240 | −18.6 | Closer approximation to the power generation natural variability allowing for more precise technical design considerations to optimize power dispatch in energy systems such as infrastructure capabilities in power generation, demand control, system synergies, sector coupling etc. |
| Pearson correlation | 0.844 | 0.8274 | +1.98 | |
| Detrended cross-correlation analysis (DCCA) coefficient | 0.793 | 0.765 | +2.74 | |
| Count [Million h] | 8.0 | | - | - |

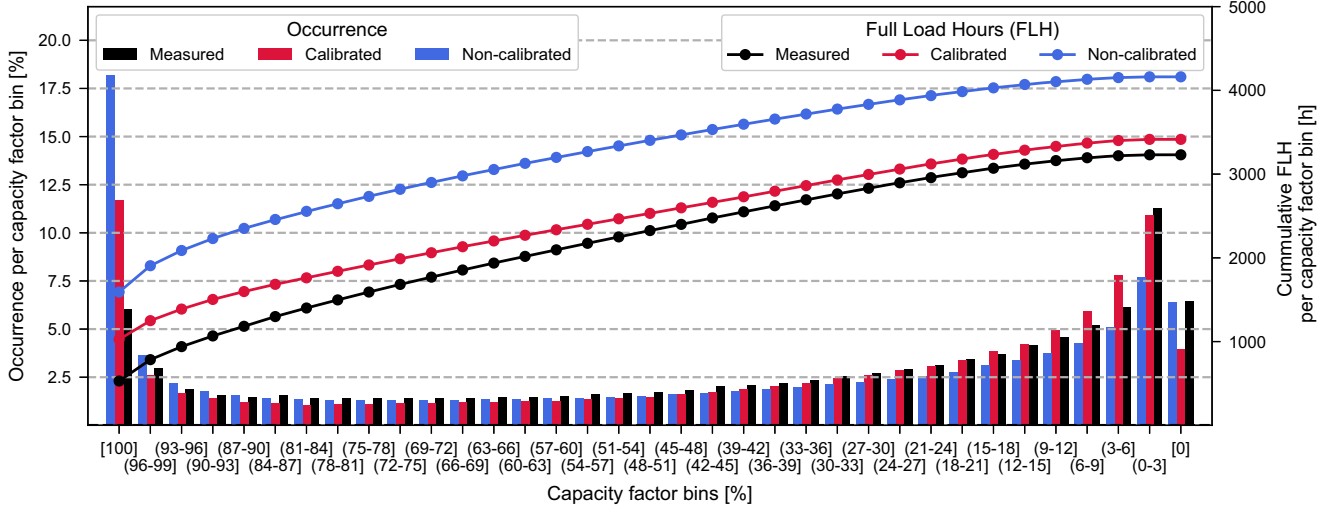

**Fig. 2 | Power generation and capacity factor bin comparison between over 8 million hourly measurements, and calibrated and non-calibrated results.** The plot illustrates capacity factor percentage bins on the x-axis, with the occurrence of capacity factors as percentages shown by bars on the left y-axis and the cumulative Full Load Hours (FLH) depicted by lines on the right y-axis. Source data are provided as a Source Data file.

adjustment, showing the correction factor peaking at approximately 12.5 m/s. At this peak, the original wind speed is adjusted downwards by about 11%. As speeds increase beyond 12.5 m/s, the magnitude of the correction gradually lessens until 17.5 m/s. The calibration effectively reverses an observed overestimation of wind speed in the range relevant for wind turbine power generation (3–25 m/s). This dependency is not possible to match with a regression of any form and highlights the relevance of a value-based wind-speed calibration approach. It is important to note that there is a steep reduction in the number of available observations at wind speeds ~20 m/s or higher, which contributes to the fluctuations seen in Fig. 1. Additionally, the measured wind speeds exhibit a skewed distribution that is well-described by a Weibull distribution, with a mean wind speed of about 6 m/s, which is a relatively low average wind speed for wind energy installations.

The impact of utilizing this wind speed calibration on the accuracy of the proposed wind power simulation workflow (cf. Method Section "Comparison with time-resolved wind turbine power generation data") is evaluated across temporal, spatial, and technological dimensions.

**Temporal dimension.** The calibration offsets over-representation of high-capacity factors in uncalibrated workflows by addressing wind speed corrections, reducing statistical errors, and ensuring closer alignment with hourly measurements from wind farms (see Table 2 and Fig. 2). The calibration procedure reduces the capacity factor mean error by 80.7% and improves temporal correlation metrics such as root mean square error, Pearson correlation, detrended cross-correlation coefficient, and Perkins' skill score (see Table 2). Despite larger deviations for higher capacity factors, the calibrated wind power workflow of *ETHOS.RESKit* achieves near-parity in total cumulative electricity generation.

Zero capacity factors occur at a similar rate (~4–7%) in both measured and simulated workflows, as wind speeds frequently fall below or exceed turbine operational thresholds (see Fig. 2). The calibration procedure has a negligible effect on these occurrences. The calibrated wind power workflow of *ETHOS.RESKit* aligns closely with measured values in the (0–3] capacity factor bin, the most frequent category. In contrast, the uncalibrated workflow underestimates occurrences by about one-third, indicating weaker temporal correlation and probability density alignment. In mid-range capacity factor bins (3–48%), the calibrated workflow aligns better with measured trends despite initially overestimating values and then declining more sharply. Conversely, in high-capacity factor bins (51–99%), the uncalibrated workflow tracks measurements more closely, though these bins contribute less to total electricity generation due to lower cumulative

occurrences. At full turbine power (100% capacity factor), both workflows significantly overestimate measurements. However, the uncalibrated workflow overshoots by 3.0 times compared to 1.9 times for the calibrated workflow, significantly impacting total generation and statistical indicators.

**Spatial dimension.** The calibrated wind power workflow of *ETHOS.RESKit* demonstrates no significant location bias across turbine types (i.e., on- or offshore) or locations when compared to both aggregated and hourly-resolved data as shown in Fig. 3, reinforcing its robustness in the spatial dimension. For hourly-resolved data, most locations show a mean capacity factor error within ±12%, with a predominantly positive deviation (overestimation, see Fig. 3)[25]. This margin is considered acceptable for generation models. However, isolated locations in Norway and Brazil show larger mean errors (-−43%), possibly due to discrepancies between simulated turbine characteristics and measurements or local GWA4 deviations.

The use of temporally aggregated generation data, which provides broader spatial coverage but lacks temporal detail, reveals a mix of trends (see Fig. 3). Wind parks in Denmark, Belgium and the USA show both negative men errors (underestimation) and positive mean errors (overestimation). Wind parks the Netherlands, on the other hand, predominantly exhibit underestimation. While useful for expanding location coverage, this approach introduces greater uncertainty due to its lack of temporal granularity.

Most mean relative errors in capacity factor by region and turbine type fall within ±14%, with the largest positive errors seen in New Zealand (+16.7%) and Germany (+25.5%) (see Supplementary Table 4). The largest negative error occurs in Brazil (−43.1%). Denmark demonstrates the most accurate results (+0.2%), followed by Belgium (+4.1%). No consistent discrepancies are linked to turbine types. Hourly-resolved data proves more reliable for precise analysis, enabling the identification of phenomena like induced stalling, restricted operation, and the exact onset of power generation. This enhances the model's ability to address spatial and operational dynamics effectively.

**Technological dimension.** In order to assess the efficacy of our model in replicating wind power generation, we conducted an experiment wherein we subjected the model to measured wind speeds at hub height. This enables the identification of potential input wind speed biases in temporal and location dependencies. However, reliable hub-height wind speed data is scarce. Only Denker and Wulf AG provided the requisite time-resolved wind speeds at hub height in conjunction with power generation from five distinct turbine models. Figure 4 compares the measurements and the simulation results obtained using the manufacturer's power curve included in the windpower.net[26] database and the synthetic power curve generator algorithm in *ETHOS.RESKit*.

A comparison of the manufacturers and synthetic power curves of Enercon and N117-2400 turbines reveals a striking similarity in shape. This is corroborated by a Perkins Skill score that is highly similar in numerical terms. This indicates that the simulated power curves are highly analogous and closely aligned with the capacity factor measurements. The line plot for the 3.4M104 Senvion shows that the simulated power curves produce significantly different sorted capacity factors compared to the manufacturer's and to the actual measurements. Possible causes of the latter might come from data handling and processing of measurements or the comparably old developing year of the turbine (2008). A newly introduced synthetic power curve score (SPCS), see Methods Section, overcomes possible data errors as well as the lack of power generation data for all turbines by directly comparing manufacturer's and synthetic power curves, bypassing the need to have time-resolved wind speeds at hub height. Table 3 presents the average SPCS for the turbines manufactured by the six

leading producers, as reported in the Windpower.net[26] database. The data in this table demonstrate that, irrespective of the wind speed input, the synthetic power curve algorithm developed[11] and included in *ETHOS.RESKit* achieves a mean power curve score of 0.96 or higher for the majority of global installed capacity. This is especially beneficial in the case where the actual power curve is unknown.

The results obtained from all three dimensions demonstrate that the *ETHOS.RESKit* wind power generation model, when used in conjunction with the calibration procedure, offers a reliable assessment tool across the different measured data classes obtained. It should be noted, however, that the availability of such data on a global scale is limited, which presents a challenge to the global validation of power simulation models.

## Evaluation against global wind power generation estimates

To address the limitation of global measurement data availability and evaluate the model's performance against global power estimates, *ETHOS.RESKit* was used to simulate historical country turbine fleets using a corrected version of the Windpower.net[26] database and compared with publicly available country level wind power estimates for several years from the International Energy Agency (IEA) (see Fig. 5 and Methods Section "Comparison with country-level statistical data"). After minimizing the effects of technology differences, temporal uncertainties, and locational variations, the model showed a very good match on global average, with national discrepancies of mostly below 10%. The IEA reports a global average capacity factor of 0.306 across 71 countries and offshore regions, while the model yielded an average of 0.287, a relative deviation of 6.2%. In comparison, the non-calibrated workflow demonstrated a significant overestimation, with an average capacity factor of 0.377 and a relative deviation of 23.1%. Regional trends are depicted in Fig. 5. At global scale, the pattern largely follows the GWA4 trend of underestimation towards and just South of the equator (cf. Supplementary Section 1.12). This also means that global correction cannot be perfect with a single set of windspeed correction factors. The small tendency for overestimation in Northern latitudes hence aligns with the findings of the location-specific comparisons at windfarm level (see above). Whilst the errors here are reduced significantly by the wind speed correction, the multinational distribution of the weather masts for calibration does perfectly reflect the specific Northern latitude trends. This in turn leads to slightly overestimated capacity factors in Northern and extreme Southern latitudes, with locally more pronounced patterns due to various reasons. Lower underestimation by GWA4 in Northwestern Europe might explain deviations in Great Britain and Germany, whilst individual outliers such as Panama, the Japanese offshore locations or Cyprus may be caused by external factors: The IEA dataset provides national energy and capacity values per year only but lacks information about individual windfarms and technology characteristics, necessitating assumptions and external data sources to define turbine properties. Further uncertainties stem from both the national and the annual averaging of generation data, which obscures spatial and temporal dynamics, and from challenges in precisely locating turbines or identifying their commissioning dates or missing design parameters. Additionally, external influences such as grid congestion, curtailment, import/export dynamics, and discrepancies in reporting conditions contribute to differences between simulated and actual results.

Calibration factors, as discussed in Supplementary Section 5.8, help mitigate these discrepancies, with national correction factors provided alongside this paper for alignment with IEA data. Furthermore, we provide raster-format correction files that extend beyond country boundaries, enabling assessments in regions without wind production and enhancing the global applicability of *ETHOS.RESKit*. This additional option in the presented workflow allows the simulation of wind turbines in any country of the world with realistic average capacity factors.

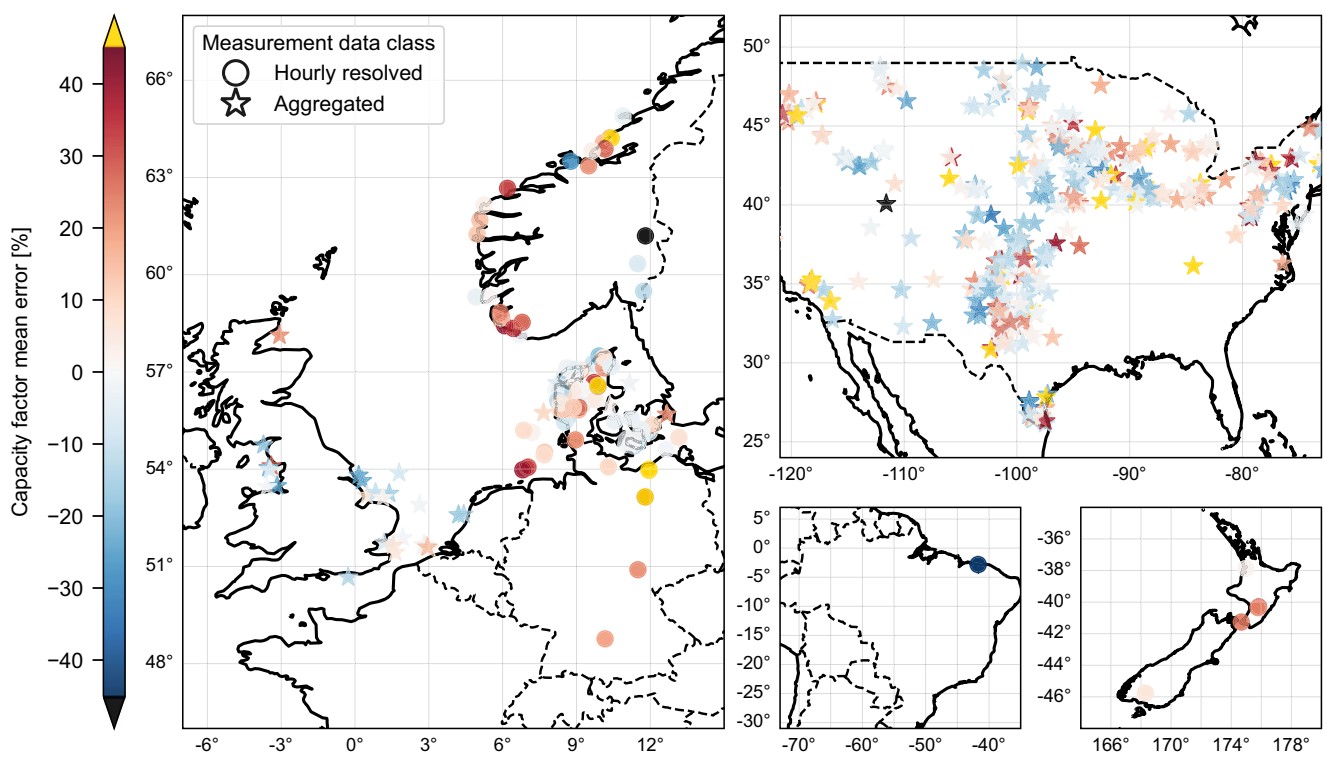

**Fig. 3 | *ETHOS.RESKit* capacity factor relative mean error comparison using two classes of obtained measurement data: hourly-resolved and aggregated.** Extracts show data regions in Europe, USA, New Zealand and Brazil (clockwise). Country shapes from GADM [25]. Source data are provided as a Source Data file.

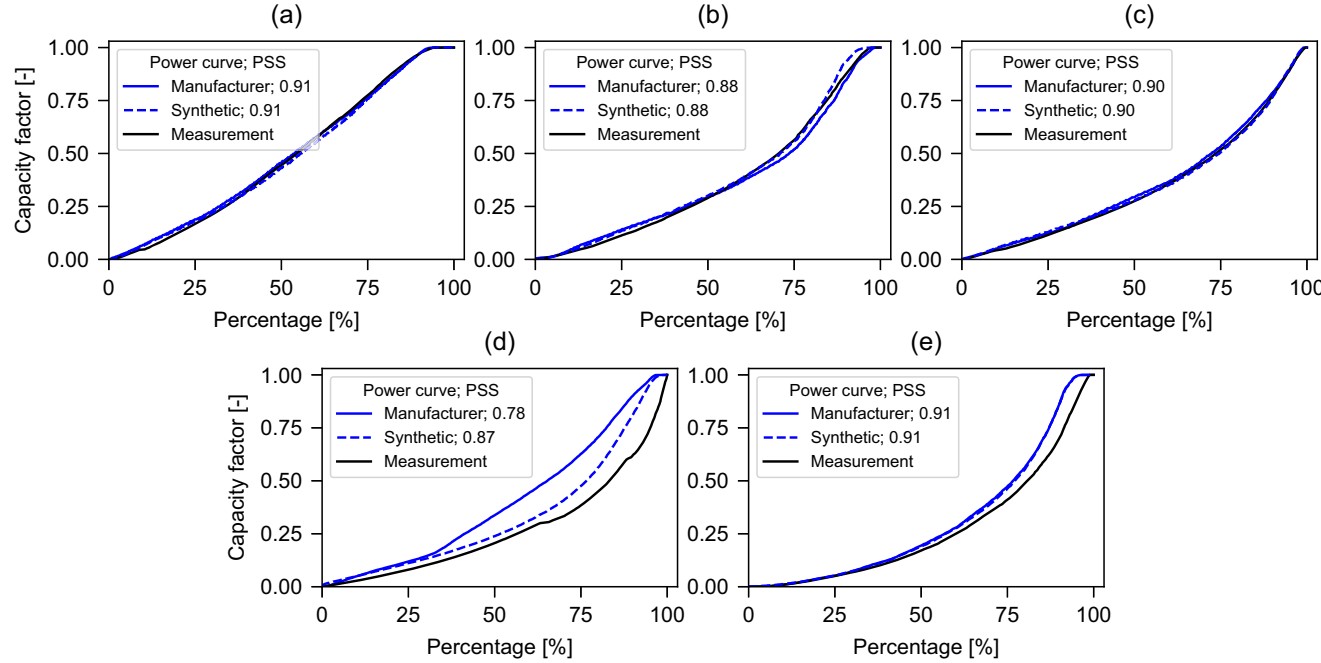

**Fig. 4 | Comparison of measured capacity factor and simulated capacity factor when using real and synthetic power curves.** Panels compare capacity factor (y-axis) as a function of turbine load, expressed as percentage of nominal power (x-axis), for five turbine models **a** Enercon E115-3000, **b** E141-4200, **c** E92-2350, **d** Senvion 3.4M104 and **e** Nordex N117-2400). Solid blue lines show simulated capacity factors derived from manufacturer power curves, and dashed blue lines show simulated capacity factors when using synthetic power curves. Black lines indicate measured capacity factor data. Perkins Skill Scores (PSS) reported in each panel quantify agreement with measured observations. Source data are provided as a Source Data file.

In conclusion, this evaluation underscores the model's ability to improve assessments of wind energy dependencies while highlighting the limitations of relying on aggregated country-level data. *ETHOS.-RESKit* demonstrates significant advancements in accuracy compared to non-calibrated workflows, setting a strong foundation for global wind energy modeling. The previously described enhancements of our model also result in superior statistical indicators in comparison to similar models such as *renewables.ninja* (see Supplementary Section 1.13).

## Discussion

In this study, we introduce an open-source, time-resolved, validated wind power simulation workflow with global applicability for the renewable energy simulation tool *ETHOS.RESKit*. *ETHOS.RESKit* leverages high-resolution wind data (250 m x 250 m) from ERA5 and GWA4, providing robust simulation capabilities and featuring the most extensive turbine model library among available tools. This library includes 880 turbine types and supports the creation of customizable synthetic power curves. The intention of this study besides describing the new wind workflow is to showcase how a methodological gap in current renewable energy simulation tools can be closed: The lack of global calibration and validation with the intention to provide reliable and highly resolved capacity factor data for wind turbines at global scale. This methodological approach developed in this work is applied to *ETHOS.RESKit* using the presented data, but is applicable also to other wind energy simulation tools.

The key innovation in the wind power workflow of *ETHOS.RESKit* therefore is its calibration process, which uses over 8 million wind speed measurements from 213 global meteorological mast sites across 25 countries. This comprehensive dataset enabled a value-dependent correction of systematic wind speed biases present in ERA5 and GWA4, ensuring improved alignment with real-world data. The calibration process significantly enhances model accuracy which is evaluated by comparing to more than 8 million hours of power generation data from 152 wind turbines across seven onshore and offshore regions. Temporal adjustments to input wind speeds shift capacity factors toward smaller values, aligning more closely with frequently measured capacity factors. This results in a 81% reduction in capacity factor deviation compared to uncalibrated simulations for turbine-level time-resolved data. When simulating historical country wind fleets, the model reduced the average capacity factor deviation to official statistics from 23 to 0.6%. Importantly, no relevant locational capacity factor trends were observed, and the model performed consistently well across both onshore and offshore regions. Furthermore, the synthetic power curve score, which evaluates alignment between synthetic and manufacturer-provided power curves, demonstrated high accuracy. Approximately 80% of globally installed turbines achieved a minimum correlation of 0.96, underscoring the precision of the model. By reducing capacity factor deviations at both the turbine and aggregated annual levels, *ETHOS.RESKit* demonstrates very good average alignment with IEA-based generation data from 71 countries. These advancements position *ETHOS.RESKit* as a leading open-source tool for global wind power modeling.

Importantly, although *ETHOS.RESKit* can simulate individual turbines, it is better suited to larger-scale assessments involving hundreds of turbine sites. Because of the spatial resolution characteristics of the ERA5 and GWA4 datasets, the model is less accurate at single locations where local wind speed conditions are not adequately represented. Furthermore, diurnal, seasonal, and terrain-based biases, as documented in the literature, fall outside the scope of our current correction method. Future work should concentrate on solving these remaining biases in order to further enhance the precision of wind power simulations. However, a significant share of these issues is contained already within the weather data, namely ERA-5 and GWA4, which is a mere input to *ETHOS.RESKit* simulation workflows. *ETHOS.RESKit* can only empirically correct the biases based on wind speed and power production observations, the reanalysis of the weather data itself, however, is out of the scope of this work and instead part of ongoing research by the producers of the respective datasets. Moreover, enhancements in higher temporal resolution and more precise local representations of wind would be advantageous for the field. Furthermore, the entire energy and climate community would greatly benefit from the availability of more publicly accessible localized time-resolved wind speeds and power generation data. In light of these considerations, the authors urge the scientific community to engage in more collaborative endeavors and to advocate for the establishment of transparent guidelines governing the accessibility of data for scientific purposes.

The findings of this study hold substantial value for the scientific and energy system analysis communities. *ETHOS.RESKit* marks a major step forward in wind power modeling, combining global applicability with high spatial resolution and the capability to simulate a wide range of technical turbine characteristics. As the first wind power simulation tool to undergo a rigorous validation and calibration process across diverse spatial and temporal scales on a global level, it sets a new standard in the field. Additionally, the inclusion of regional correction factors enhances the precision of wind energy assessments, even in areas currently lacking wind turbine installations. By enabling more accurate simulations, the open-source tool equips decision-makers with critical insights to optimize renewable energy utilization and make strategic investments. This advancement significantly supports the integration of renewable energy into global power systems.

**Table 3 | Average synthetic power curve score in *ETHOS***

| Manufacturer | Global installed capacity [GW] | Percentage of global capacity | Turbines installed [thousand] | Synthetic power curve score[1] |
|---|---|---|---|---|
| Vestas | 110.6 | 29.68 | 49.7 | 0.988 |
| Enercon | 45.4 | 12.19 | 24.2 | 0.965 |
| GE Energy | 43.0 | 11.54 | 25.0 | 0.984 |
| Siemens | 38.9 | 10.43 | 14.2 | 0.985 |
| Gamesa | 36.0 | 9.67 | 24.5 | 0.994 |
| Nordex | 21.7 | 5.81 | 8.7 | 0.992 |
| Total | 295.6 | 79.32 | 146.3 | 0.984 |

[1]the synthetic power curve score is the cumulative minimum sum of capacity factors distribution of two power curves: manufacturer and synthetic, taking as reference the manufacturer one. *RESKit* for the turbines of the top six manufacturers according to the installed capacity reported according to windpower.net[26].

## Methods

In this section, we outline the methodology employed for our wind power simulation and validation approach implemented in the wind power workflow of *ETHOS.RESKit*[11,27] (see more details about *ETHOS.RESKit* in Supplementary Section 1.6), aimed at providing a basis for global wind energy assessments. As visualized in Fig. 6, the methodology is structured into four subsections covering (a) data acquisition and processing, (b) deriving global wind speed calibration factors aiming at addressing potential mean errors in the underlying weather data, (c) a subsequent extensive validation of our wind power simulation workflow by comparing against time-resolved park level power generation data, country-level power generation data and national statistical data, and (d) deriving national correction factors. Each step ensures the accuracy and robustness of the employed simulation framework.

### Data Acquisition, Classification, and Processing

In the following, we will address the acquisition, classification, and processing of data crucial for the validation and enhancement of our wind power simulation workflow. The data sources encompass global wind speed measurements, wind turbine power generation records, information on existing windfarms, and historical national wind electricity power generation data and are listed in Supplementary Table 4. Each dataset plays a distinct role in refining our simulation model, either through correction or validation processes.

### Wind speed measurement data

To initiate the study, we collected 18.3 million hourly, mostly openly available recordings from 1980 to 2022 of wind speeds from

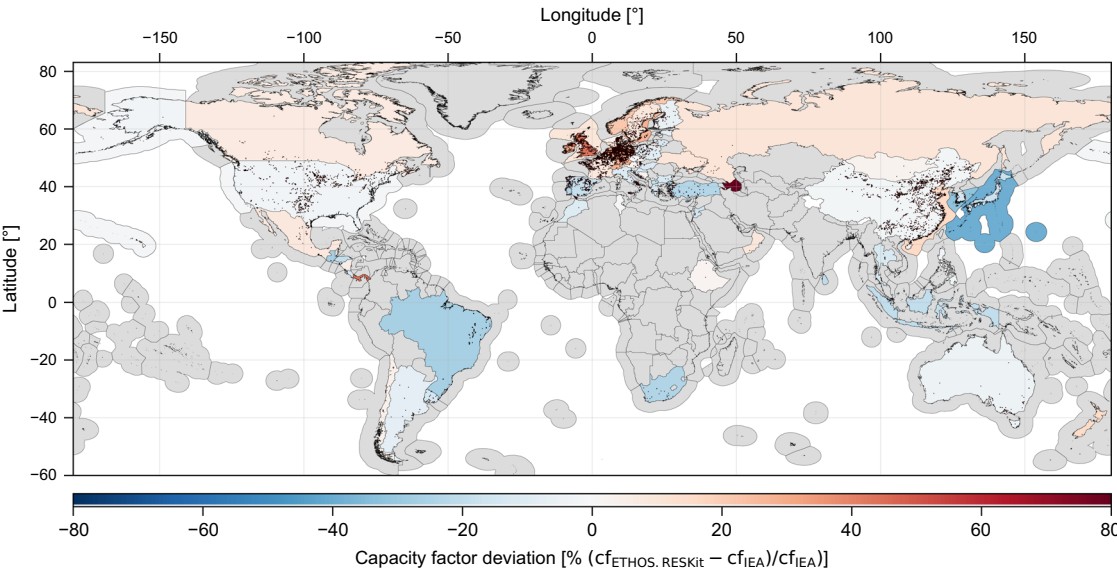

**Fig. 5 | Capacity factor deviation map between *ETHOS.RESKit* and IEA data.** Global map showing the average relative deviation in country-level capacity factor (cf) between *ETHOS.RESKit* simulations and IEA data [24] for the years 2017 to 2021. Colors indicate the relative deviation in capacity factor $(\mathrm{cf_{ETHOS.RESKit}} - \mathrm{cf_{IEA}})/\mathrm{cf_{IEA}}$ in percentage. Blue shading denotes where *ETHOS.RESKit* underestimates IEA values, and red shading denotes overestimation. Gray areas correspond to countries without available data. Country shapes from GADM [25]. Source data are provided as a Source Data file.

meteorological masts worldwide, ranging in height from 40 to 160 m at 210 locations in 25 countries. These recordings are utilized to derive a wind speed correction. Measurements from masts at ground level (10 m) have only been compared to simulated data from GWA4[15] to qualitatively detect potential global biases in ERA5 (cf. Supplementary Fig. 1) and justify the detailed regional correction approach. The data has not been included into the quantitative calibration and validation methodology though since relevant wind speed heights for turbine simulations are around 100 m and large height differences entail additional sources of error[28]. We nevertheless hope to contribute to creating awareness for potential biases within quasi-standard wind datasets such as the Global Wind Atlas or ERA-5. Measurements were harmonized following the steps outlined in Section 1.2 of the Supplementary. Utilizing quality control information provided together with the measured wind speeds (cf. Supplementary Section 1.3), we filtered out erroneous measurements, e.g., no valid recording, negatives, duplicated values, etc. For further processing, we resampled the measured wind speeds and those from ERA5 to hourly values, standardized them to UTC time, and saved their geolocations as well as the measurement heights respectively within one netCDF file per measurement height. All used input data sources are listed in the Supplementary Table 4.

### Wind turbine electricity power generation data
A total of 8 million hourly recordings of turbine electricity generation from 152 onshore and offshore wind turbines and wind farms from 2002 to 2021 from 6 countries globally were collected from various, mostly proprietary, data sources and will be employed in a validation of our wind turbine simulation workflow. Harmonizing this data involved a process analogous to the wind speeds procedure and involved converting the power output time series to a capacity factor time series by dividing the measured power with the nominal capacity. Furthermore, in the case of wind farm data, the reported electricity output was converted into a capacity factor time series by dividing by the total park capacity.

As a quality control measure, we applied an algorithm to filter out out-of-normal operations such as curtailment, maintenance, or other irregularities from the gathered data to avoid distorting the validation results. For this, we simulated the capacity factors of the respective turbines (see Supplementary Section 1.6) to first exclude observation periods in which the measured capacity factor was zero while the simulated capacity factor was greater than 0.4 to account for erroneous measurements. Second, we filtered observation periods in which the measured capacity factor exhibited zero for longer than a day to capture maintenance. Lastly, we filtered values where the measured capacity factor does not change for a minimum of 5 h, while the difference between the measured and simulated capacity factor is greater than 0.1 to filter out curtailment lasting longer than 5 h. Lastly, the processed data was saved in one netCDF file per location.

### Database of existing wind farms
Additionally, we acquired a proprietary database on existing wind farms containing data on 26,900 wind farm locations worldwide as well as databases on turbine models and power-curves available from thewindpower.net[26] to simulate the existing wind fleet stock and derive national correction factors. The databases include, for instance, information on geolocation, capacity, number of turbines, hub height, turbine model, commissioning, and decommissioning dates until July 2022. It furthermore includes a turbine model database with data on the manufacturer, rated power, rotor diameter, market introduction, and minimum and maximum available hub heights of turbine models. To harmonize and check these databases, preprocessing, void-filling to estimate missing values and data filtering steps are performed as outlined in Supplementary Fig. 6. Furthermore, for some entries, erroneous data was identified by manual examination. The manual examination for example involved countries with few wind farms where the capacity and capacity development of the entries in the wind farms database differed substantially from the capacity reported by the IEA Renewable Energy Progress Tracker[29]. If found to be erroneous, data on location, capacity and commissioning dates were manually corrected using additional sources such as reports, OpenStreetMap and satellite data, if possible (s. Supplementary Section 1.5 for more details). These modifications are documented and provided as Supplementary Data 3.

Finally, we removed locations with turbine capacities lower than 1 MW, as such turbines are comparably old and typically exhibit very low hub heights, leading to unrealistic simulation outcomes in *ETHOS.RESKit*, which is specifically designed for potential assessments

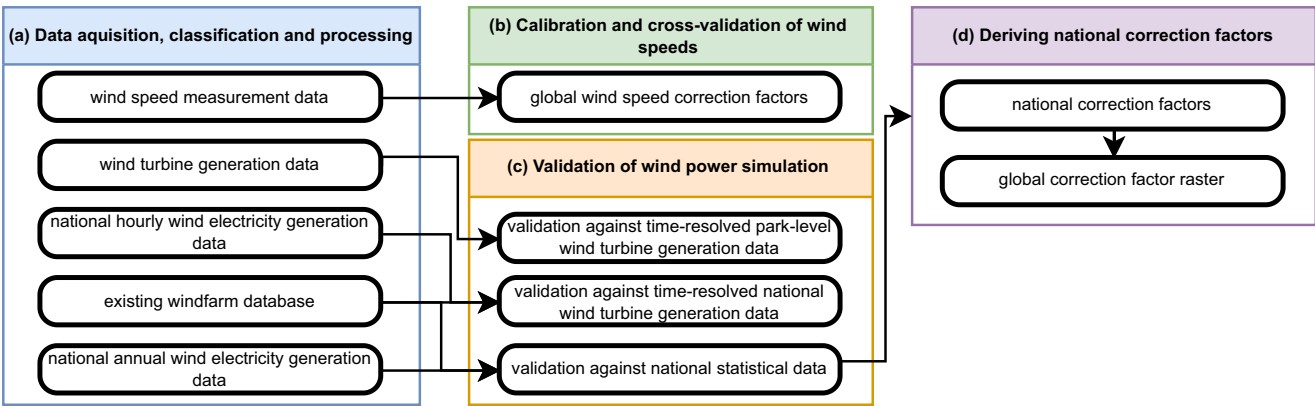

**Fig. 6 | Overview of the applied methodological steps. a** Data acquisition, classification and processing of datasets including wind speed measurements, wind turbine generation data, national wind electricity generation data and wind farm database. **b** Calibration and cross-validation of wind speeds to derive wind speed correction factors. **c** Validation of wind power simulations against park-level and national time-resolved data and national statistical data. **d** Derivation of national correction factors and generation of a global correction factor raster. Panel colors denote workflow stages: blue (**a**), green (**b**), orange (**c**) and purple (**d**).

of future energy systems. This arises from the substantial downscaling distance required from the 100 m ERA5 wind speed height to the turbine hub-height, introducing inherent uncertainties in the wind-speed values. In this context, such wind turbines with small hub-heights and low capacity are anticipated to have a marginal impact on the total power generation of a country due to their small capacity, justifying their exclusion from the analysis.

### Country-level statistics and time series data

We obtained annual wind power generation and capacity data from 2017 to 2021 for 71 countries and offshore regions from the IEA Renewable Energy Progress Tracker[29] as a basis for calculating national capacity factors to derive national correction factors for our simulation workflow. Data prior to 2017 has not been included as there was limited global installation of wind capacity in those years and average electricity yields are distorted by a high proportion of older, smaller turbine models. To avoid distortions in capacity factors due to capacity additions during a year, a capacity-weighted capacity factor considering monthly or even daily capacity additions was derived. This sub-annual factor was based on commissioning dates from the employed wind farm database and an extensive manual search to correct and complement the database as well as the IEA data (see Supplementary Section 1.5). In Equation), index $i$ denotes the respective wind farm, while ophours is the number of hours the wind farm was operational in the respective year based on the commissioning date, and IEA and WD (wind farm database) indicate the data source.

$$\text{cf}^{\text{IEA, weighted}}_{\text{country, year}} = \frac{\text{gen}^{\text{IEA}}_{\text{country, year}}}{\text{cap}^{\text{IEA}}_{\text{country, year}}} * \frac{1}{\frac{\sum_{i,\text{ country}}(\text{ophours}^{\text{WD}}_{i,\text{ country}} * \text{cap}^{\text{WD}}_{i,\text{ country}})}{\text{cap}^{\text{WD}}_{\text{country, year}}}} \quad (1)$$

The weighted capacity factor is especially necessary for countries with limited wind turbine capacities or a large share of commissioned capacity within a year as small deviations in the data have a large impact on the reliability of the calculated capacity factor and therefore the validation results.

In summary, Fig. 7 shows the type and locations of the real-world data that were considered within this study. Statistical country values are available for various countries across the globe with data gaps predominantly in Africa, South America and South Asia. Weather mast measurements are available mainly from the USA, Europe, South-Africa and Iran while wind farm measurements are limited to the North-Sea area and Norway.

### Calibration and cross-validation of estimated wind speeds from reanalysis weather data

We used the hourly measured wind speeds from meteorological masts to employ a calibration and cross-validation of the reanalysis wind speeds from ERA5 (including GWA4-downscaling) to correct for mean errors and overall under- or overestimations in the wind speed values reported by several publications[17,19,20,30]. For the calibration and cross-validation we focused on wind speeds above 2 m/s due to the operational range of wind turbines[21,24] and measurement heights between 40 and 160 m, resulting in 8.4 million hourly measurements. In a first step, we extracted the wind speeds processed within *ETHOS.RESKit* for the same locations, heights, and time periods of the weather masts without applying wake losses or any other correction factors (see Supplementary Section 1.6 for a detailed description). In a second step, wind speeds were binned in 0.1 m/s categories and a proportional regression per bin was used to fit processed and measured wind speeds. A weighing based on the number of mast-specific hours with measurement data was selected as it yielded the most even global distribution. Individual regressions per global region, landcover type and latitude or slope bin proved infeasible due to the limited amount of available measurement data at global scale. Comparative analyses showed similar trends for various world regions, however, which supports the validity of an averaged global regression function here. Alternative regressors were also tested but discarded as our tests indicated signs of overfitting or worse performance (see Supplementary Section 1.7.1).

The applied proportional regression function is given in Equation) and is defined by a scaling factor $a$ per wind speed bin. The proportional regressor underwent fitting and validation through k-fold cross-validation. For this the data was split into 210 folds, with each fold corresponding to a mast, with the goal of assigning equal weight to each mast. The scikit-learn Python library[31] was utilized for performing the k-fold split. The choice of k-fold cross-validation is motivated by its suitability for our methodology, considering that other approaches such as the leave-one-out approach proved computationally intensive, and a rolling cross-validation performed worse than the k-fold cross-validation during initial testing.

The cross-validation procedure results in 210 fitted regressors, subsequently averaged into a single regressor from which a single scaling factor $a$ per wind speed bin is extracted. These factors were then used to correct the wind speeds within *ETHOS.RESKit* according to Equation),

$$\text{ws}_{\text{corr}} = a(\text{ws}_{\text{raw}}) * \text{ws}_{\text{raw}} \quad (2)$$

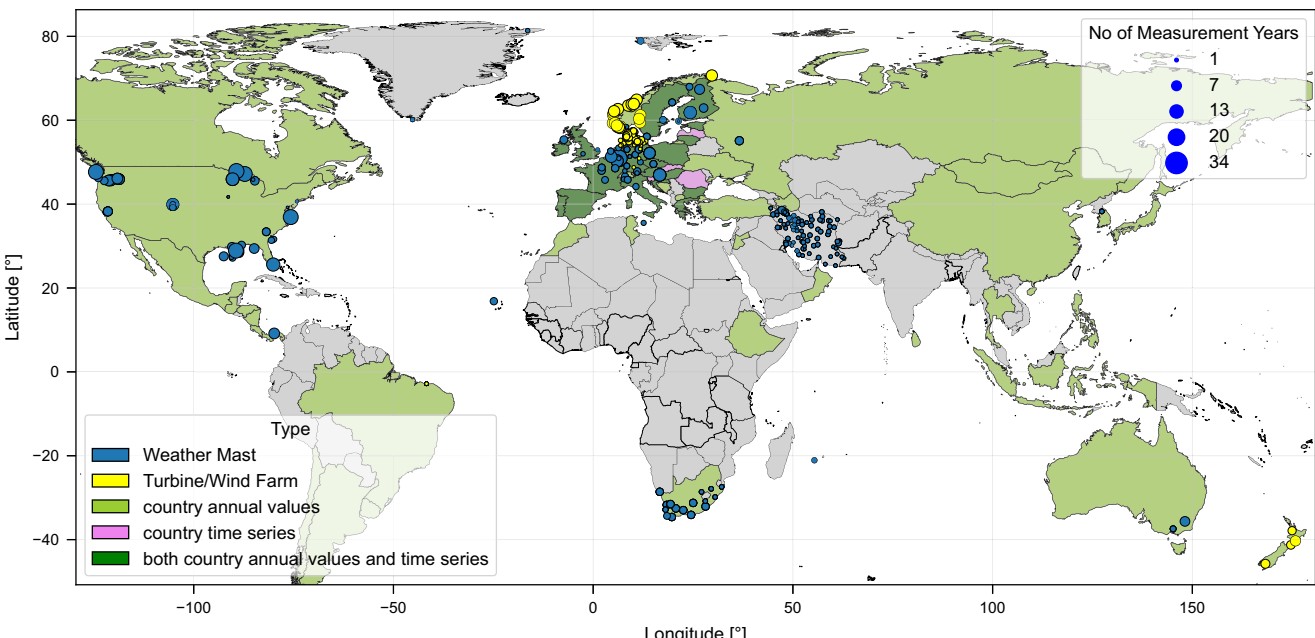

**Fig. 7 | Spatial overview of location and type of real-world data considered within this study.** Data on specific locations, such as measurement data of weather masts (yellow) and wind turbines or wind farms (blue) are shown as circles. The size of the circles indicates the number of measurement years per location. Data on country level such as annual country capacity factors (light green), aggregated country time series (pink) or both (dark green) are indicated by coloring the country respectively. Source data are provided as a Source Data file.

where $ws_{corr}$ represents the corrected wind speed, and $ws_{raw}$ denotes the uncalibrated modeled wind speed. This unified calibration aims to rectify any general under- or overestimation present in the data. The resulting wind speed dependent scaling factors can be found in Fig. 1b and the supplementary files. This wind speed correction is applied to every location simulated within *ETHOS.RESKit*. The wind speed correction factors are provided as Supplementary Data 1.

To assess the quality of the regressor, we utilized a scoring function, using the mean error (ME) to account both for general deviation as well as over- or underestimation of wind speeds and capacity factors. We computed various metrics common in the literature[18,20,30,32,33] to assess the quality of the cross-validation procedure and evaluate our results. To assess the temporal correlation between measured and simulated time series, we evaluated the Pearson correlation and the detrended cross-correlation analysis (DCCA) coefficient. In addition, we used the Perkins' skill score (PSS), a probability density function, to evaluate the normal distribution. Moreover, we proposed a new probability density function called synthetic power curve score SPCS, based on the PSS from 0 to 1, where 1 represents an exact match, with the difference that it uses the cumulative minimum capacity factor distribution of two power curves, taking as reference the power curve of the manufacturer. The SPCS is described in Equation) where ws is the wind speed in each location at hub height, *Capacity factor* is the respective capacity factor distribution corresponding to wind speed ws for the manufacturer's and the synthetic power curve respectively.

$$SPCS = \sum_{0}^{ws} \min\left( Capacity\ factor_{manuf,ws}, Capacity\ factor_{synth,ws} \right) \quad (3)$$

Further analysis involves evaluating diurnal and seasonal mean errors in simulated wind speeds and reanalysis data. Results are given in Supplementary Section 1.7 as the main focus of this study is the *ETHOS.RESKit* wind power simulation workflow.

## Comparison with time-resolved wind turbine power generation data

Next, we validated the employed *ETHOS.RESKit* wind power simulation workflow by comparing it to processed hourly measured turbine power generation data. First, *ETHOS.RESKit* was utilized to simulate the wind turbine power generation time-series for the measured time spans of each real turbine considering their specific hub heights, rotor diameters, and real power curves, if available. Those power curves are based on the information provided by the manufacturers and may be considered theoretical compared to the measured power curve of an installed turbine on site. In cases where real manufacturer power curves were unavailable, a synthetic power curve was generated based on the specific power and a fit derived from 130 manufacturer power curves[11]. Simulations were executed both with and without applied wind speed correction to assess the potential improvements in the simulation workflow. Furthermore, wind speed losses due to wake effects are considered using the wind efficiency curve (knorr-mean) from windpowerlib[34]. These wake losses were also considered for all turbine simulations in the results. Electrical and mechanical turbine losses are considered inherently via the wind speed dependent $C_p$ factor within the power curves[35], downtimes related e.g. to maintenance are considered by an additional availability factor of 0.98[36,37]. In the preprocessing step maintenance times were already filtered out. Subsequently, we assessed the difference in results using various metrics, including root mean square error, DCCA coefficient, and relative mean error. These assessments occurred at the location level. Aggregated assessments are calculated by weighing each location equally when calculating metrics.

## Comparison with country-level statistical data

As the regional coverage of the available time-resolved wind turbine power generation data is limited, and our workflow is intended for global use, we first further validated and subsequently calibrated our model by comparing it against annual turbine and wind farm level power generation output and country-level annual capacity factors

from the IEA[29]. The years 2017 to 2021 were used since previous years saw only limited growth of wind capacity.

Annual turbine and wind farm level power generation output from turbines in the United States, Denmark and the United Kingdom were used additionally as they are publicly available. For each location, the average reported capacity factor was calculated using capacity and power generation. Afterwards, the locations were simulated within *ETHOS.RESKit* and the simulated capacity factor was compared with the reported capacity factor.

The filtered wind turbine database, with missing data filled in was first used to simulate country-level capacity factors by applying the method described. Second, the database was used to derive IEA-based country-level capacity factors by accounting for intra-annual capacity additions. Extensive data checks with the following data exclusion rules have been applied: If less than 75% of the official IEA capacity is reported in the wind farm database, the corresponding year was discarded, as this indicates that the wind farm database is incomplete for that year. Omitting this year would potentially lead to large discrepancies, as a different wind fleet would be simulated compared to the one that existed in that year. Additionally, we excluded years in which the country's IEA capacity was less than or equal to 3 MW, as such a low capacity suggests a limited number of plants, where small errors in the input data could result in significant deviations in the simulated country's capacity factor. Furthermore, we discarded a country or the respective year of that country if too many entries in the wind farm database are deemed erroneous. Therefore, the number of considered years varies for each country. The list of the final countries and years considered can be found in Supplementary Section 1.8. Finally, we validated the performance of our simulation workflow on a global scale by comparing the resulting, simulated annual country-level capacity factors against IEA-based country-level capacity factors by calculating the average deviation in capacity factors for every year.

Furthermore, to be able to correct our simulation workflow towards official country statistics, we derived additional correction factors, which can be optionally applied in *ETHOS.RESKit*. For this, we calculated a capacity-factor correction factor for every country, representing the average deviation in capacity factors between the IEA-based country-level capacity factors and our simulated country-level capacity factors over the years 2017 to 2021 according to Equation):

$$f_{country}^{corr} = \text{mean}\left(\frac{cf_{country, year}^{IEA}}{cf_{country, year}^{RESKit}}\right). \tag{4}$$

This inverse average deviation served as a country correction factor ($f_{country}^{corr}$) implemented in *ETHOS.RESKit* to correct the electricity output. To avoid capacity factors above 1 and retain load peaks, the electricity output was corrected by adjusting the processed wind speed instead of directly correcting the simulated capacity factor. This wind speed adjusting is performed iteratively until the capacity factors match with a tolerance of 1%.

Not all countries worldwide can be covered with this approach as only a limited number of countries have installed relevant wind farm capacities. We assume that the observed deviations mostly stem from regional mean errors from which neighboring countries are also affected. Therefore, we derive a global raster of correction factors provided as Supplementary Data 4, enabling the application of global correction at any point in the world. The global raster is created by assigning every wind farm location used in this study to the respective country correction factor value and applying a global spatial interpolation over these locations. This way, existing regional mean errors are also corrected in countries without any current wind farm capacities.

## Data availability
Wind speed correction factors, country capacity factor correction factors, modifications to the thewindpower.net database and the IEA data and the global raster of capacify factor correction factors generated in this study are provided as Supplementary Data 1, 2, 3 and 4, respectively. Source data are provided with this paper. The wind farm database was obtained under a commercial license from thewindpower.net and cannot be publicly shared due to licensing restrictions. Access to the database can be obtained by purchasing a license directly from thewindpower.net[26]. The time-resolved wind-turbine power-measurement data used in this study are available under restricted access because they were obtained from third-party providers under individual agreements. Access can be obtained by inquiring directly with the respective providers listed in Supplementary Table 4. The raw measurement data are protected and are not available due to data-ownership restrictions. Further data sources used in this study are listed in Supplementary Table 4. Source data are provided with this paper.

## Code availability
The model is freely available on the institute's GitHub page (https://github.com/FZJ-IEK3-VSA/RESKit). Additionally, the source code of the *ETHOS.RESKit* model used in this study has been archived at Zenodo[27].

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

## Acknowledgements

We acknowledge the following people and organizations for providing dataset used in this study. We are grateful for their contribution to this work. Henning Weisbarth - Denker & Wulf AG. A major part of this work has been carried out within the framework of the H2 Atlas-Africa project (03EW0001) funded by the German Federal Ministry of Education and Research (BMBF) (authors: EUPS, CW). Part of this work has been carried out within the framework of the HyUSPRe project which has received funding from the Fuel Cells and Hydrogen 2 Joint Undertaking (now Clean Hydrogen Partnership) under grant agreement No 101006632. This Joint Undertaking receives support from the European Union's Horizon 2020 research and innovation program, Hydrogen Europe and Hydrogen Europe Research (authors: PD). This work was partly funded by the European Union (ERC, MATERIALIZE, 101076649). Views and opinions expressed are, however, those of the authors only and do not necessarily reflect those of European Union or the European Research Council Executive Agency. Neither the European Union nor the granting authority can be held responsible for them (authors: HH). This work was supported by the Helmholtz Association under the program "Energy System Design" (authors: EUPS, PD, CW, HH, FP, JW, RM, SD, TK, JL, DS). European Union's Horizon 2020 research and innovation program (grant agreement No. 758149) (authors: KG). Open Access Publications funded by the Deutsche Forschungsgemeinschaft (DFG, German Research Foundation) – 491111487.

## Author contributions

Conceptualization: E.U.P.S., P.D., C.W., and H.H.; methodology: E.U.P.S., P.D., F.P., C.W., H.H.; software: E.U.P.S., P.D., and C.W.; validation: E.U.P.S., P.D., F.P., and C.W.; formal analysis: E.U.P.S., P.D., and C.W.; investigation: E.U.P.S., P.D., C.W., H.H., J.W., R.M., S.D., and S.C.; data curation: E.U.P.S., P.D., C.W., F.P., R.M., S.D., S.C., J.W., and K.G.; writing – original draft: E.U.P.S., P.D., and H.H.; writing – review and editing: E.U.P.S., P.D., C.W., H.H., T.K., J.W., J.L., and S.C.; visualization: E.U.P.S., P.D., and H.H.; supervision: H.H., J.L., and D.S.; project administration: H.H.; funding acquisition: H.H. All authors have read and agreed to the published version of the manuscript.

## Funding

## Competing interests

The authors declare no competing interests.

## Additional information

[1]Institute of Climate and Energy Systems - Jülich Systems Analysis (ICE-2), Forschungszentrum Jülich GmbH, Jülich, Germany. [2]Chair for Fuel Cells, Faculty of Mechanical Engineering, RWTH Aachen University, Aachen, Germany. [3]Chair for Energy Systems Analysis, Department of Mechanical Engineering, University of Siegen, Siegen, Germany. [4]Institute of Bio- and Geosciences - Agrosphere (IBG-3), Forschungszentrum Jülich GmbH, Jülich, Germany. [5]Institute for Sustainable Economic Development, University of Natural Resources and Life Sciences, Vienna, Austria. [6]These authors contributed equally: E. U. Peña-Sánchez, P. Dunkel, C. Winkler. ✉e-mail: p.dunkel@fz-juelich.de

