## [Transparent Peer Review file · Nature Communications]

Towards high resolution, validated and open global wind power assessments

Corresponding Author: Mr Philipp Dunkel

Version 0:

Reviewer comments:

Reviewer #1

(Remarks to the Author)

General comments

The paper presents an interesting analysis to provide global time series of turbine performance. One can say that Fig. 2-5 and Table 1 and 2 are all validation of the model. The new method (applying the calibration as a function of wind speed), is represented mostly in Fig. 1. The balance between methods and validations seems off, i.e. there is too much focus on the validation and too little on new methods. So in Fig. 1 (and in the paper in general) the main point is to calibrate the model as a function of wind speed, but this will be hard to generalize to other model chains and is therefore not so interesting for the readers. I would have liked the model description more up front and in more detail, so that it becomes easier to understand the paper. It was very hard for me to puzzle together the different pieces, perhaps this is partially a consequence of the nature format with a lot of supplementary materials, but in general cross-references to where a certain topic was discussed could be improved (see detailed comments below).

It is great that the performance is compared with existing models, but the comparison is only based on 21 (or 22?) measurements points. I can see a lot of new datasets and quality control of the data has been performed, so I am wondering if some data-journal might be a better fit for this interesting work. Also in some cases the explanations are not clear and I think it would be hard to reproduce the results in the paper (for example the specific comments about Sec. 1.6). In addition, abbreviations are not always defined, which makes it hard to understand for outsiders of this specific domain. For example, NUTS2, ENTSO-E, MERRA2, ERA5, HadISD. So based on these combined problems I am unfortunately unable to recommend this paper for publication.

Specific comments

I25: metrerological -> meteorological

I45: I think it would be better to talk here about "Extractable wind power resources" or something similar, because wind power resources itself don't depend on the wind turbine technology. For example, the commonly used 'wind power density' just indicates how much energy is available in the wind.

I65: What kind of validation are you talking about here? I think that would helpful to add at the beginning of this paragraph.

I97-199: Not sure if I follow this: in the introduction you are saying that Pygreta and Reskit use a scaling from the GWA3. Isn't that the same a wind speed correction measure?

I103: "the available number varies" -> "the available types of turbines varies"

I130-142: I am quite confused what is being calibrated here and how. Just scaling the wind speeds up and down based on wind speeds doesn't really provide any insights in the physical process that play a role. It could be attributed to many things in the model chain and is therefore hard to generalize. Most importantly information on the height is lacking here: if everything is validated with 10 m masts, it is not clear what contributes to these observed biases. And scaling based on those values may not necessarily improve hub height winds. Generally information on the model chain that is used is missing here or not presented in a clear way (it is not possible for a reader to understand the relation between section 1.6 and the results shown here).

I345: This should contain a reference where the quality assurance procedure is defined.

I348: it would be useful to include a direct link to the measurements dataset that has been made after this procedure.

I367: Also here a direct link to a dataset would be useful.

I471: For me it seems like the the main focus in this paper is the validation.

Comments on supplementary materials:

Section 1.1:

It is confusing that in I342 you state that 10 m masts have not been included, but the first result in the supplementary materials is 10 m wind speeds.

Section 1.2

ROOT MEAN SQUARE ERROR: why in capitals?

Section 1.6:

- 1) You use the word "downscaling" here, but isn't downsampling more appropriate?
 - 2) downscaled -> for me this implies that some information has been added to get to result at higher resolution, but that seems not to be the case here. You have just resampled the data a higher resolution until this step?
 - 3) Why do you divide the LRA by the GWA3 here? It makes more sense to use the actual period that the GWA3 was run for, otherwise there will be a lot of random noise due to year-to-year variations? Also the 40 years mentioned here do not correspond with Fig. 7 where you talk about a 10 year period.
 - 4) It is confusing for me to use the word "Projected" here, because that is usually used for the maps or coordinate systems. I would say vertically extrapolating/interpolating is more appropriate. However, information is missing here how this logarithmic interpolation is done. Also simple log interpolation will fail to capture time-varying stability effects in the atmosphere.
 - 5) What kind of correction?
 - 8) How do you adjust for air density? Which elevation data do you use for this correction?
- Fig. 7: this is partially hard to read because the text does not fit in the boxes.

Section 1.9:

This is an interesting comparison, but the methods are not clearly described. What does "country-level hourly power generation" mean, is that one aggregated number for the whole country? What is ENTSO-E?

Section 1.10:

The last part describes the effect that larger turbines were better described than low turbines: how did you determine that these were the result of power curve representation and terrain effects. I can understand why the production close to the ground is harder to predict, because the models do not resolve small scale features at these heights, but why would the power curve be less accurate for smaller turbines?

Section 1.13:

22 locations: this is very little. Could you indicate if these results are statistically significant? The text should point to the data that are included in this comparison, otherwise it should not be included. Fig 10 and Table 2: why is there 22 data points in Fig 10 and 21 in Table 2? Where is this comparison performed on the github examples you provided?

(Remarks on code availability)

Reviewer #2

(Remarks to the Author)

What are the noteworthy results?

- The introduction of the validated tool ETHOS.RESKitWind for global wind power generation simulation, including a library with 880 turbine types as well as support for the creation of customisable synthetic power curves.
- The innovative calibration process, which uses over 8 million wind speed measurements from global meteorological mast sites across 25 countries and more than 8 million hours of power generation data from 152 wind turbines across seven onshore and offshore regions.

Will the work be of significance to the field and related fields? How does it compare to the established literature? If the work is not original, please provide relevant references.

- The developed tool improves on existing global wind energy simulation tools by including regional calibration factors.

Does the work support the conclusions and claims, or is additional evidence needed?

- Yes, the claims are backed up with the results.

Are there any flaws in the data analysis, interpretation and conclusions? Do these prohibit publication or require revision?

- The difference between wind power and wind energy should be discussed, as I feel it is not used consistently. You talk a lot about power, but capacity factor is a reflection of the energy production.
- In this paper, the developed tool is stated to be transparent, open source, validated and evaluated. The relevance and need for an "open source" tool is not discussed. Please explain this in the paper.

Is the methodology sound? Does the work meet the expected standards in your field?

- The methodology is generally sound and meets the expected standards in the field.
- Please include a discussion about if a national split is really the best way to split the data. Did you look at other factors, such as terrain type?
- Please explain how you accounted for electrical losses, which are present in measurement data but not in the simulations? I assume these are part of the calibration, but maybe there is some variation over wind speed, turbine type, location, wind farm size, etc.
- Please include a description of which database technology you used, how it was implemented, and how the data was structured and managed.
- On line 477, you refer to "real power curves". Can you please clarify if this means "real theoretical power curves" or "real

measured power curves”? I suspect it’s the first option, in which case you should comment on the fact that the actual site-specific power curves may vary from the theoretical power curves. Also, please explain how turbine-specific synthetic power curves were generated.

Is there enough detail provided in the methods for the work to be reproduced?

• Yes, and the Python toolkit is available on GitHub. I haven’t tried to use it, though.

Comments on the abstract:

• The abstract needs to be more specific:

o It’s not clear what you mean exactly with “Wind power simulations to support deployment strategies vary drastically in their results, hindering reliable design decisions.” Which “wind power simulations” are you referring to? And with “design”, are you referring to wind turbine design or wind farm design or both? These are two very different things. Please rephrase to take account of this.

o It’s not clear what the need for a transparent, open source, validated and evaluated, global wind power simulation tool is. Please explain this.

o Please clarify what the “global wind power simulation tool” actually simulates.

o Please quantify how high the spatial resolution is. And are the 22 customizable designs for wind turbines or for wind farms?

o Please clarify the statement “We achieve a global average capacity factor mean error of 0.006 and Pearson correlation of 0.865.”. Does this refer to a comparison to energy yield measurements? Over what time periods? Where were all the projects? This is difficult to understand, and the abstract should be understandable independently of the rest of the paper.

(Remarks on code availability)

Reviewer #3

(Remarks to the Author)

The paper presents a transparent, open source, validated and evaluated, global wind power simulation tool called ETHOS.RESKitWind with high spatial resolution and customizable designs for both onshore and offshore wind turbines. The following comments are for the authors of the paper.

The paper did not specify the types of wind turbines that the authors claimed they simulated in the abstract and introduction. According to the authors 800 wind turbines were simulated, and which type of analysis did they carry out during the simulation?

The paper is not written as a standard Journal paper. From Introduction then Results, no models, no theory, no descriptions of control to help bring out the methodology and technical aspects of the paper.

The paper is scattered and not well structured. Method employed is coming after Results.

Besides, there is no typical model of the wind farm with detailed parameters and ratings for easy replication of the study.

The presented results are limited and not showing the performance of some of the key variables of the wind turbines employed.

(Remarks on code availability)

The code looks fine.

Version 1:

Reviewer comments:

Reviewer #1

(Remarks to the Author)

General comments

The paper is relevant and outlines a method to obtain capacity factors for energy planning purposes. There is a lot of information in the paper, although the structure is partially still confusing due to the nature format with supplementary materials. The code is nicely documented and the authors have done a great job adding a toolchain to achieve many things.

The paper is improved, but there is still methodological problems. In step 4 of the supplementary materials (I226) you are using the local roughness length to extrapolate around hub height. This is not valid in the majority of onshore cases. At 10 m height, using the local roughness length may work quite well, but this becomes very problematic at 100 m, because the local roughness length is often irrelevant, because the footprint at 100 m is typically several kilometers upstream of the current wind direction. For example, we are near the coast and the local roughness is 1 m (forest), but the wind is coming 95% from offshore directions. In that case the roughness length should be close to 0.0002 m and not 1 m! The proper way to do this is to use the mesoscale roughness length and speedup-factors due to internal boundary layers, but this may be beyond the scope of the paper. An alternative is to revise this extrapolation method and use the observed wind shear from the different

heights of the global wind atlas. But it is well documented that vertical extrapolation does not follow the log-law at greater heights (e.g. <https://link.springer.com/article/10.1007/s10546-007-9166-9>).

The global wind atlas 4 has been released (<https://doi.org/10.11583/DTU.28955267>), so in a way the paper contains a outdated dataset. Ideally it would be nice to redo the results using this improved version, but can understand if that is a lot of work. So if this is not possible, it would be good to comment on this in the introduction or discussion. In any case I would probably notify the reader which Global Wind Atlas version is used by writing for example GWA3 instead of GWA.

I am still confused by the calibration process. Is the correction factor defined by country? Or is there one for the whole world? If the latter, the correction factors are entirely defined by the countries which are presented in the tall tower dataset. What happens if you leave out countries like Iran, for which there is many towers used in calibration. Wouldn't this totally change the correction factors? In other words: without more justification for this wind speed dependent tuning it is hard to defend that you can apply them anywhere in the world. They are fully determined by the masts used in the calibration and thus not general.

Minor comments:

l348 "the resolution of" -> "solving"

l374: "comprehensive" -> better to leave these subjective words out

l377: "robust" -> better to leave these subjective words out

l517: found in the Supplementary -> where (Table/Fig) exactly?

Suppl materials l438: "as shown in (...?)"

(Remarks on code availability)

Very nicely documented code with many examples.

Reviewer #2

(Remarks to the Author)

Thank you for responding to my comments so thoroughly - I am now happy with the changes and would like to accept the paper for publication

(Remarks on code availability)

Reviewer #3

(Remarks to the Author)

Despite the revision, the authors still presented the results of the paper before the methods.

(Remarks on code availability)

Despite the revision, the authors still presented the results of the paper before the methods.

Version 2:

Reviewer comments:

Reviewer #1

(Remarks to the Author)

General comments:

The manuscript is improved and the main issues have been addressed. There is still a lot of ad-hoc decisions in the model chain that I would have probably done differently, but that is unavoidable in a model chain that encompasses so many fields (wind resource modelling, wake modelling, power system). I would recommend the authors to look at the reference below for more inspiration.

I have some minor comments that still need to be addressed:

Main article:

l36: remove "the most". I would argue there is other methods that are equally comprehensive (e.g. <https://doi.org/10.1016/j.apenergy.2021.117794>).

l181: skewed normal distribution -> it is for sure not normal but rather a weibull distribution because the wind speed can never be lower than 0.

Supplementary materials:

1.2:

Figures Figure 2, Figure 3, Figure 4, Figure 5 -> Figures 1,2,3,4 and 5

1.3:

throttling -> curtailment

1.6

bullet 1: Long-run average -> Long-term average is more frequently used

bullet 3: long-term orographic -> long-term microscale (the GWA contains more than just orographic speedups, but also

those due to roughness/stability).

bullet 4: which levels? I would call this: Vertical extrapolation of wind speed

1.12

Fehler -> fix ref

1.12 more than 67% -> you mean the ratio is lower than 0.67?

(Remarks on code availability)

Dear editor and reviewers of the manuscript “Towards high resolution, validated and open global wind power assessments” with ID: NCOMMS-24-84206. We thank you for your time and your constructive comments to improve the manuscript. We confirm that we have addressed each comment in the revised manuscript as outlined below.

We have structured our answers in a tabular format such that first your questions are listed in blue, in the green box below we answer your question and summarize the resulting changes from the manuscript and supplementary material finally in the red box below. Any reference to line numbers in the replies below refers to the originally submitted version from the first review to avoid confusion.

Reviewer #1 (Remarks to the Author):

1	The paper presents an interesting analysis to provide global time series of turbine performance. One can say that Fig. 2-5 and Table 1 and 2 are all validation of the model. The new method (applying the calibration as a function of wind speed), is represented mostly in Fig. 1. The balance between methods and validations seems off, i.e. there is too much focus on the validation and too little on new methods.
Response	Thank you. We indeed focused on the validation part more than usual since it represents a crucial aspect of our approach, filling a methodological gap of many existing tools and studies: It ensures the global reliability and accuracy which is a core element of our workflow and results. Furthermore, it represents a methodological step in itself which can be applied also to other wind energy simulation workflows or tools at global or regional scale. We have added additional sentences in the introduction that highlight the importance of model validation.
Changes	In line 53: A methodological gap persists, however, when it comes to the calibration and validation of such wind power and energy simulation models. For example, a previous article [8] found that only 21% of studies assessing large-scale wind resource potentials conduct a validation of the input data. For the open-source simulation tools, the situation is even worse: Whilst Renewables.ninja provides a calibration over selected European countries [9], no wind energy simulation tool is validated at global scale to the knowledge of the authors. The consequence is a lack in reliability of the simulation results. The present paper addresses this shortcoming by developing a global calibration and validation process which is demonstrated

	using the example of the open-source ETHOS.RESKit but is applicable also to other wind energy simulation tools.
2	So in Fig. 1 (and in the paper in general) the main point is to calibrate the model as a function of wind speed, but this will be hard to generalize to other model chains and is therefore not so interesting for the readers. I would have liked the model description more up front and in more detail, so that it becomes easier to understand the paper. It was very hard for me to puzzle together the different pieces, perhaps this is partially a consequence of the nature format with a lot of supplementary materials, but in general cross-references to where a certain topic was discussed could be improved (see detailed comments below).
Response	Thank you. We have to adhere to Nature's guidelines. We agree that the paper would be easier to understand if the Methods section came before the Results section. We have now added cross-references throughout the paper to improve navigability. While the actual factors of the wind speed correction that are applied are indeed specific to our proposed workflow, our calibration methodology can generally be applied to other wind energy simulation tools, since the necessary input data is publicly accessible. Furthermore, our tool is open source. This means that the workflow, and therefore the calibration, can be used by anyone. We have clarified this with additional statements in the Introduction and Discussion part, please see below.
Changes	In line 53: [...] The present paper addresses this shortcoming by developing a global calibration and validation process which is demonstrated using the example of the open-source ETHOS.RESKit but is applicable also to other wind energy simulation tools. In line 274: The intention of this study besides describing the new wind workflow is to showcase how a methodological gap in current renewable energy simulation tools can be closed: The lack of global calibration and validation with the intention to provide reliable and highly resolved capacity factor data for wind turbines at global scale. This methodological approach developed here is applied to ETHOS.RESKit in the present publication, based on the presented data, but is applicable also to other wind energy simulation tools. In line 131: The calibration of input wind speeds as described (c.f. Methods section "Calibration and cross-validation of estimated wind speeds from reanalysis weather data") In line 143: The impact of utilizing this wind speed calibration on the accuracy of the proposed wind power simulation workflow (c.f. Method section "Comparison with time-resolved wind turbine power generation data") is evaluated across temporal, spatial, and technological dimensions.

	In line 234: [...] International Energy Agency (IEA) (see Figure 5 and Methods section “Comparison with country-level statistical data”).
3	It is great that the performance is compared with existing models, but the comparison is only based on 21 (or 22?) measurements points.
Response	Thank you. The low number (22) of turbine locations can be explained by our very strict selection of turbines for the model comparison. We only selected real turbine locations which were a) covered geographically by all selected models in the comparison and b) for which the exact turbine model was available in the model database. We had initially considered to also use similar turbine models or synthetic approximations but have decided against doing so to avoid distortions. We understand that this should be explained in more detail and have added below explanation.
Changes	Appendix: It is imperative to acknowledge that not all locations and measured times could be simulated with each tool by the authors, and only 22 locations were common amongst all the three models. A wider range of locations would have been possible if deviations from the exact turbine model or even synthetic power curves had been accepted, however, the exact match has been selected here to exclude potential biases against models with lesser turbine model coverage.
4	I can see a lot of new datasets and quality control of the data has been performed, so I am wondering if some data-journal might be a better fit for this interesting work.
Response	Thank you. The focus is rather on the methodological gap that all current renewable energy simulation tools have: None of those is calibrated/validated at global scale, i.e. we can never be sure how reliable the simulation results are, especially when simulating non-European placements. This problem is addressed methodologically in our paper, using the example of ETHOS.RESKit , which is presented and published open source for other users. The data collection and processing is a very central element of this workflow, but only meant to support the before-mentioned goal. The core methodology that we develop in the paper is applied to ETHOS.RESKit exemplarily, but can be applied to any wind energy simulation tool. We have clarified this now in the introduction and discussion, see below adaptations.
Changes	In line 273: This library includes 880 turbine types and supports the creation of customizable synthetic power curves. The intention of this study besides describing the new wind workflow is to showcase how a methodological gap in current renewable energy simulation tools can be closed: The lack of global calibration and validation with the intention to provide reliable and highly resolved capacity factor data for wind turbines at global scale. This methodological approach developed here is applied to ETHOS.RESKit in the

	present publication, based on the presented data, but is applicable also to other wind energy simulation tools. The key innovation in the wind power workflow of ETHOS.RESKit therefore is its calibration process, ...
5	Also in some cases the explanations are not clear and I think it would be hard to reproduce the results in the paper (for example the specific comments about Sec. 1.6).
Response	Thank you. We have adapted several sections with workflow step descriptions and have made them more specific in order to ease reproducibility. Please have a look at for example section 1.6 that you mention, but also other parts (often found in the supplementary materials due to the detailed nature of these steps but partly also the main text). Below your example of section 1.6 in the supplementary material:
Changes	 1. Resampling: ERA5 wind speeds at a height of 100 m are resampled to match the grid spacing of GWA3 using linear interpolation. 2. Long-run average (LRA): A ten-year average (2008-2017, matching the exact years that were also used for the generation of GWA3 [9]) is calculated based on the hourly ERA5 data at the desired location. 3. Correction factor: The LRA is divided by the value from GWA3, resulting in a correction factor. This factor is then applied to the resampled ERA5 time series data to improve the representation of long-term orographic effects. 4. Wind speed height effect: The corrected wind speeds are extrapolated to the hub height (or anemometer height) according to the logarithmic wind profile [10], which has been widely used for this task in the literature [11][12]. The surface roughness length used in the formula varies with land-use and land-cover retrieved from the Climate Change Initiative of the European Space Agency (ESA) [13] and are available in the ETHOS.RESKit package as one of multiple default settings for the logarithmic scaling function input. Planetary boundary layer effects are also considered by only applying wind speed scaling when the target height is lower than or equal to the top of the boundary layer. 5. If applicable, a wind speed correction is employed using wind-speed dependent regressors as described in the section “Calibration and cross-validation of estimated wind speeds from reanalysis weather data” of the main document (see also Section Error! Reference source not found. below for background information). 6. If applicable, wind speed losses due to wake effects are considered using the wind efficiency curves from windpowerlib[14]. 7. If available, the manufacturer’s power curve is used, otherwise a synthetic power curve, as described in Ryberg et al. [7], is applied.

	8. To calculate the power output at the turbine location, the following steps are carried out: a) Air density correction: The simulated wind speed is adjusted for air density based on the IEC 61499-12-1:2017 standard [15]. The air density correction is resolved hourly and based on actual wind speed, pressure at ground level, temperature, with the turbine height and sea level reference and the following constants:  1) $g_0 = 9.80665$ Gravitational acceleration [m/s²], 2) $M_a = 0.0289644$ Molar mass of dry air [kg/mol], 3) $R = 8.3144598$ Universal gas constant, [N·m/(mol·K)], 4) $\rho_{STD} = 1.225$ Standard air density [kg/m³]. b) Power curve convolution: The adjusted wind speed is convolved with the power curve using a scaling factor of 0.01 and a base factor of 0. c) Power output simulation: The power curve is applied to the simulated wind speeds to simulate the power output or capacity factors. d) If applicable, a power-output correction factor is employed (e.g. country correction factor or losses) that further corrects wind speeds to meet target capacity factor.
6	In addition, abbreviations are not always defined, which makes it hard to understand for outsiders of this specific domain. For example, NUTS2, ENTSO-E, MERRA2, ERA5, HadISD. So based on these combined problems I am unfortunately unable to recommend this paper for publication.
Response	Thank you. We have now added the full names of the respective datasets, terms etc. in parenthesis whenever the term is first introduced. For datasets, we have additionally linked to the reference which also contains a URL.
Changes	In line 55: “MERRA-2 (Modern-Era Retrospective Analysis for Research and Applications v2) [13]” In line 57: “ERA5 (European Centre for Medium-Range Weather Forecasts Reanalysis v5) [15] data” In line 73: “ENTSO-E (European Network of Transmission System Operators for Electricity) [24]” In the appendix: “HadISD (Hadley Integrated Surface Database)” In the appendix: “NUTS2 (Nomenclature of Territorial Units for Stistics)”
7	l25: metrerological -> meteorological
Response	Thank you.
Cha	The typo has now been corrected.

8	l45: I think it would be better to talk here about "Extractable wind power resources" or something similar, because wind power resources itself don't depend on the wind turbine technology. For example, the commonly used 'wind power density' just indicates how much energy is available in the wind.
Response	Thank you. We have adapted our wording accordingly.
Changes	In line 45: Thus, evaluating extractable wind energy resources [...]. In line 50: Extractable wind energy resources depend on the location (spatial dependency), on [...].
9	l65: What kind of validation are you talking about here? I think that would helpful to add at the beginning of this paragraph.
Respo	Thank you. We have added the following sentence:
Changes	In line 68: Validation, as understood by the authors, involves comparing model outcomes with real-world data to assess how accurately the simulation represents actual observed conditions.
10	l97-l99: Not sure if I follow this: in the introduction you are saying that Pygreta and Reskit use a scaling from the GWA3. Isn't that the same a wind speed correction measure?
Response	Thank you. You are correct. pygreta (at least in some workflows) also employs GWA for a simple scaling of local wind speeds, similar to how it is done in ETHOS.RESkit. Doing this already improves a) the spatial resolution and b) accuracy significantly, but the results still contain considerable bias and inaccuracies. The scope of the present work therefore exceeds this by far by correcting the biases introduced by ERA-5 and GWA via calibration/validation measures. This means introducing additional correction factors which depend (among others) on the actual hourly wind speed as well as the geographical location of the turbine. Only this subsequent step ensures accurate simulation results with high reliability as it is needed for energy system design in policy making. We have highlighted this more prominently in introduction and discussion now, see below changes.
Changes	In line 53: A methodological gap persists, however, when it comes to the calibration and validation of such wind power and energy simulation models. For example, a previous article [8] found that only 21% of studies assessing large-scale wind resource potentials conduct a validation of the input data. For the open-source simulation tools, the situation is even worse: Whilst Renewables.ninja provides a calibration over selected European countries [9], no wind energy simulation tool is validated at global scale to the knowledge of the authors. The consequence is a lack in reliability of the simulation results.

	The present paper addresses this shortcoming by developing a global calibration and validation process which is demonstrated using the example of the open-source ETHOS.RESKit but is applicable also to other wind energy simulation tools.
11	I103: "the available number varies" -> "the available types of turbines varies"
Response	Thank you. The old phrase has been replaced to add clarity.
Changes	In line 102: As presented in Error! Reference source not found. , most models offer to simulate the performance of such turbines although the available types of turbines vary from 27 to 141.
12	I130-142: I am quite confused what is being calibrated here and how. Just scaling the wind speeds up and down based on wind speeds doesn't really provide any insights in the physical process that play a role. It could be attributed to many things in the model chain and is therefore hard to generalize.
Response	Thank you. You are right, we cannot assess the underlying physical drivers that cause deviations in the wind speed data since we "start later in the process": ERA-5 and Global Wind Atlas wind speed data is considered "input data" in our workflow. The reanalysis of the weather data itself is out of our scope and possibilities, it would, however, be possible for the authors of the Global Wind Atlas or ERA-5 to fix the observed biases and deviations already in an improved reanalysis process with a stronger focus on validation. The data available to us (reanalysis data from ERA-5/GWA vs. measured wind speed and wind farm power output data) allows only for an empirical approach: We compare the hourly simulated wind speeds against measured wind speed data at hourly resolution, deriving different calibration factors for different wind speeds and then apply an overall scaling based on rolling averages of real wind energy production vs. simulated data. This has been added also in our discussion. First glimpses at possible underlying issues can be found in the analysis of the GWA deviations though (see supplementary material), where we are able to show that the GWA deviations correlate with e.g. certain latitudes, land cover types and geography. These effects do not suffice though to correct electricity generation output sufficiently at global scale, therefore we developed the real-wind-energy-generation-data based post-correction step which takes geographical differences into account.
Changes	In line 299: However, a significant share of these issues is contained already within the weather data, namely ERA-5 and GWA, which is a mere input to ETHOS.RESKit simulation workflows. ETHOS.RESKit can only empirically correct the biases based on wind speed and power production observations,

	the reanalysis of the weather data itself, however, is out of the scope of this work and instead part of ongoing research by the producers of the respective datasets.
13	Most importantly information on the height is lacking here: if everything is validated with 10 m masts, it is not clear what contributes to these observed biases. And scaling based on those values may not necessarily improve hub height winds. Generally information on the model chain that is used is missing here or not presented in a clear way (it is not possible for a reader to understand the relation between section 1.6 and the results shown here).
Response	Thank you. This is indeed a source of a potential misunderstanding. The 10m data is not used for the validation for the exact reasons that you are pointing out, only weather masts close to the actual hub heights (between 40-160m measurement height) are used. The 10m data is used only to qualitatively assess the global biases and indicatively show that/why the regional assessment is necessary. We have clarified that in the “Wind speed measurement data” section now. Furthermore, we have added cross references to the respective Methodology sections, where the steps that were performed are outlined. There, we also reference Supplementary section 1.6 where the simulation workflow is described.
Changes	In line 342: Measurements from masts at ground level (10m) have only been compared to simulated data from GWA [14] to qualitatively detect potential global biases in ERA5 (cf. Figure 1 in the Supplementary Material) and justify the detailed regional correction approach. The data has not been included into the quantitative calibration and validation methodology though since relevant wind speed heights for turbine simulations are around 100m and large height differences entail additional sources of error [27]. We nevertheless hope to contribute to creating awareness for potential biases within quasi-standard wind datasets such as the Global Wind Atlas or ERA-5. In line 131: The calibration of input wind speeds as described (c.f. Methods section “Calibration and cross-validation of estimated wind speeds from reanalysis weather data”) [...].
14	l345: This should contain a reference where the quality assurance procedure is defined.
Response	Thank you. The reference to the respective appendix section, where the quality indicators are listed and explained, has been added.

Changes	In line 344: Utilizing quality control information provided together with the measured wind speeds (cf. Supplementary Section 1.3), [...]
15	I348: it would be useful to include a direct link to the measurements dataset that has been made after this procedure.
Response	Thank you. All datasets for the wind speed measurements except one are publicly available. The sources are listed in the Supplementary Table 4.  1. Ramon J, Lledó L, Pérez-Zanón N, Soret A, Doblas-Reyes FJ. The Tall Tower Dataset: a unique initiative to boost wind energy research. Earth System Science Data 2020;12:429–39. https://doi.org/10.5194/essd-12-429-2020  - Downloadable via: https://talltowers.bsc.es/access-the-data 2. Kubistin D, Plaß-Dülmer C, Arnold S, Kneuer T, Lindauer M, Müller-Williams J, et al. ICOS Atmosphere Level 2 data, Steinkimmen, release 2023-1 2023. https://doi.org/10.18160/BJ1Z-BE0T  - Downloadable via: https://data.icos-cp.eu/portal/ 3. Santos, Pedro; Mann, Jakob; Vasiljevic, Nikola; Courtney, Michael; Sanz Rodrigo, Javier; Cantero, Elena; et al. (2019). The Alaiz Experiment (ALEX17): wind field and turbulent fluxes in a large-scale and complex topography with synoptic forcing. Technical University of Denmark. Collection. https://doi.org/10.11583/DTU.c.4508597.v1  - https://data.dtu.dk/articles/dataset/ALEX17_Multi-lidar_measurements_vizualizing_flow_patterns_with_2D_and_3D_wind_fields_over_complex_terrain/7931444?file=27242726 The missing dataset belongs to the meteorology department of the Jülich Research Centre Safety and Radiation Protection institute. We have initiated a process to hopefully soon see the data public as well.
Changes	In line 348: All used input data sources are listed in the Supplementary Table 4.
16	I367: Also here a direct link to a dataset would be useful.
Response	Thank you. We regret to inform that no open dataset with sufficient coverage and level of detail exists for our global yet very detailed assessment, forcing us to use proprietary data. The data that is referred to in line 367 is still at turbine/wind farm level and therefore allows to draw inferences about the original data from thewindpower.net. We can therefore not share this dataset publicly and hope for your understanding. The data can of course be reproduced exactly though with the exact steps that we described, once one has access to thewindpower.net data which is commercially available to everyone. The dataset that was used can be found under this link: https://www.thewindpower.net/store_continent_en.php?id_zone=1000

Changes	In line 547: [...] and are listed in Supplementary Table 4.
17	l471: For me it seems like the main focus in this paper is the validation.
Response	Thank you. We would partly agree: Even if the paper contains more novelties (such as first and foremost the new and openly available workflow, but also observations on existing biases in global wind speed and energy distribution), the validation is one core element that differentiates the ETHOS.RESKit workflow presented here from other workflows which lack a global calibration/validation and therefore pose challenges with regard to reliability of wind simulation results at global scale. The focus is therefore on the methodological gap that causes a lack of reliability and accuracy especially when simulating wind power outputs at global scale. The calibration and subsequent validation are the methodological remedy to this problem then, and at the same time the observed deviations prove the necessity for a new, properly calibrated workflow as we present it here. Furthermore, the calibration methodology is applicable to other wind energy simulation workflows as well, allowing for more calibrated/validated simulation tools in the future.
Changes	In line 53: A methodological gap persists, however, when it comes to the calibration and validation of such wind power and energy simulation models. For example, a previous article [8] found that only 21% of studies assessing large-scale wind resource potentials conduct a validation of the input data. For the open-source simulation tools, the situation is even worse: Whilst Renewables.ninja provides a calibration over selected European countries [9], no wind energy simulation tool is validated at global scale to the knowledge of the authors. The consequence is a lack in reliability of the simulation results. The present paper addresses this shortcoming by developing a global calibration and validation process which is demonstrated using the example of the open-source ETHOS.RESKit but is applicable also to other wind energy simulation tools.
(The following comments apply to the appendix)	
12	Section 1.1: It is confusing that in l342 you state that 10 m masts have not been included, but the first result in the Supplementary is 10 m wind speeds.
Response	Thank you. This was indeed confusing but should now hopefully be clear with the adapted paragraph below.
Changes	In line 342: Measurements from masts at ground level (10m) have only been compared to simulated data from GWA [14] to qualitatively detect

	potential global biases in ERA5 (cf. Figure 1 in SM) and justify the detailed regional correction approach. The data has not been included into the quantitative calibration and validation methodology though since relevant wind speed heights for turbine simulations are around 100m and large height differences entail additional sources of error [27].
13	Section 1.2: ROOT MEAN SQUARE ERROR: why in capitals?
Response	Thank you. It was supposed to be the abbreviation “RMSE” to explain the variable name on the y-axis of Fig. 5 a little later. The term has been adapted to RMSE.
Changes	“...coefficient, root mean square error (RMSE), mean ...”
14	Section 1.6: 1) You use the word "downscaling" here, but isn't downsampling more appropriate?
Response	Thank you. The sentence was indeed hard to read, so we decided to rephrase the whole sentence to express more clearly what the challenge is here. We avoid the terms “downscaling” or “downsampling” altogether now and instead use the well-established term of “logarithmic scaling” for scaling wind speeds to different heights, which is what we employ. Additionally, we changed “downsampling” in 1.6 1) to “resampling” (c.f. the next answer).
Changes	In the Supplementary: “This is primarily due to the uncertainty in the wind speeds at hub height, which results from the large scaling factors when wind speeds are scaled logarithmically from 100m height (as provided in the ERA5 data) to very low hub heights.” Section: 1.6 1) Resampling: ERA5 wind speeds at a height of 100 m are resampled to match the grid spacing of GWA3 using linear interpolation.
15	Section 1.6: 2) downscaled -> for me this implies that some information has been added to get to result at higher resolution, but that seems not to be the case here. You have just resampled the data a higher resolution until this step?
Response	Thank you. You are correct, “resampled” would have been the better term here to express what we did and is used now in the latest version of bullet points 1 and 3. In bullet point 2, the term is actually not even necessary for the understanding and has been removed for conciseness. Thank you.
☺ ☺	1. Resampling: ERA5 wind speeds at a height of 100 m are resampled to match the grid spacing of GWA3 using linear interpolation.

	2. Long-run average (LRA): A forty-two-year average (1980-2022) is calculated based on the downscaled hourly ERA5 data at the desired location. 3. Correction factor: The value from GWA3 is divided by the value the LRA, resulting in a correction factor. This factor is then applied to the resampled ERA5 time series data to improve the representation of long-term orographic effects.
16	Section 1.6: 3) Why do you divide the LRA by the GWA3 here? It makes more sense to use the actual period that the GWA3 was run for, otherwise there will be a lot of random noise due to year-to-year variations? Also the 40 years mentioned here do not correspond with Fig. 7 where you talk about a 10 year period.
Response	Thank you. We indeed had the same thought which evolved over the course of our work: after starting with the long-run average of all available years, we decided to change this to only the years that were used for generating the Global Wind Atlas interannual averages (which is 2008-2017, see https://globalwindatlas.info/en/about/method). Hence the (correct) 10-years in Fig. 7 – the other “forty-two-year average (1980-2022)” is a remainder of the previous approach. Also, we indeed divide the GWA by the LRA. This was a mistake in the text.
Changes	2. Long-run average (LRA): A ten-year average (2008-2017, matching the exact years that were also used for the generation of GWA3 [9]) is calculated based on the hourly ERA5 data at the desired location. 3. Correction factor: The value from GWA3 is divided by the value of the LRA, resulting in a correction factor.
17	Section 1.6: 4) It is confusing for me to use the word "Projected" here, because that is usually used for the maps or coordinate systems. I would say vertically extrapolating/interpolating is more appropriate. However, information is missing here how this logarithmic interpolation is done. Also simple log interpolation will fail to capture time-varying stability effects in the atmosphere.
Response	Thank you. It is true that fluctuations under instable atmospheric conditions cannot be represented with the chosen logarithmic scaling method. This is a simplification that we had to make for the sake of data size and calculation efficiency in the workflow, which is intended also for large-scale calculations over multiple years. We do consider boundary layer effects though. The wording of the sentence has been adapted to better explain what we did, avoiding the term “projection/projected” and using the term “logarithmic scaling” now. Also, references to the applied methodology for the scaling and the data source have been added. The extracted roughness factors over CCI landcover class are available in the RESkit package as one of multiple default settings for the logarithmic scaling function input.

Changes	Wind speed height effect: The corrected wind speeds are extrapolated to the hub height (or anemometer height) according to the logarithmic wind profile [10], which has been widely used for this task in the literature [11][12]. The surface roughness length used in the formula varies with land-use and land-cover retrieved from the Climate Change Initiative of European Space Agency (ESA) [13] and are available in the ETHOS.RESkit package as one of multiple default settings for the logarithmic scaling function input. Planetary boundary layer effects are also considered by only applying wind speed scaling when the target height is lower than or equal to the top of the boundary layer.
18	Section 1.6: 5) What kind of correction?
Response	Thank you. The wind speed range is subdivided into wind speed bins, and every of these bins receives its own linear wind speed-dependent regressor which is confirmed by an additional k-fold cross-validation. This binning approach allows us to address and fix the wind speed error in our simulation which we found to be (on geographical average) inconsistent across the wind speed range. The procedure is explained in detail in the section “Calibration and cross-validation of estimated wind speeds from reanalysis weather data” of the paper. This reference, however, was missing so far in this bullet point of the appendix, we have now added the cross-reference to the main document as well as to the additional comparison with other regressor approaches.
Changes	If applicable, a wind speed correction is employed using wind-speed dependent regressors as described in the section “Calibration and cross-validation of estimated wind speeds from reanalysis weather data” of the main document (see also Section Error! Reference source not found. below for background information).
19	Section 1.6: 8) How do you adjust for air density? Which elevation data do you use for this correction?
Response	Thank you. The air density is corrected at hourly resolution using the IEC 61499-12-1:2017 standard, based on actual wind speed, pressure at ground level, temperature, with the turbine height and sea level reference and the following constants: $g_0 = 9.80665$ Gravitational acceleration [m/s ²], $M_a = 0.0289644$ Molar mass of dry air [kg/mol], $R = 8.3144598$ Universal gas constant, [N·m/(mol·K)], $\rho_{STD} = 1.225$ Standard air density [kg/m ³]. Please feel free to have a look at the actual function here: https://github.com/FZJ-IEK3-VSA/RESKit/blob/39eb9225fcb813d0fa5870e24db799602bd2e9dc/reskit/wind/core/air_density_adjustment.py#L4 The reference to the IEC standard has been added.
Changes	In the Supplementary: The simulated wind speed is adjusted for air density based on the IEC 61499-12-1:2017 standard [15]. The air density correction

	is resolved hourly and based on actual wind speed, pressure at ground level, temperature, with the turbine height and sea level reference and the following constants:  1) $g_0 = 9.80665$ Gravitational acceleration [m/s²], 2) $M_a = 0.0289644$ Molar mass of dry air [kg/mol], 3) $R = 8.3144598$ Universal gas constant, [N·m/(mol·K)], 4) $\rho_{STD} = 1.225$ Standard air density [kg/m³].
20	Section 1.6: Fig. 7: this is partially hard to read because the text does not fit in the boxes.
Response	Thank you. You are right, please excuse. In addition to the font sizes (and boxes) being too small, somehow also resolution has been reduced. We have now restructured the whole figure (using a snake-like flow pattern now) and managed to increase the readability a lot with larger fonts and boxes.
Changes	Please see replaced Figure 7.
21	Section 1.9: This is an interesting comparison, but the methods are not clearly described. What does "country-level hourly power generation" mean, is that one aggregated number for the whole country?
Response	Thank you. Country-level hourly power generation means that there is one hourly generation time-series per country. Sometimes ENTSO-E provides the hourly generation at a sub-national level. In this case, we aggregated the time series to national level. We describe the methodology for this in section 1.5 of the Appendix. We added a reference to Section 1.5 to the text.
Changes	Figure 8 compares country-level hourly power generation data from ENTSO-E and our simulations for 2017-2021 using the DCCA coefficient to assess correlation. The methodology is described in section 1.5. In Section 1.5: ENTSO-E provides absolute hourly generation time-series at national or sub-national level. In case of sub-national data, the time-series were aggregated to national level through summation.
22	Section 1.9: What is ENTSO-E?
Response	Thank you. ENTSO-E stands for "European Network of Transmission System Operators for Electricity". We have introduced the term properly now (as well as other abbreviations that we have used) at the beginning of the publication, together with the respective references wherever applicable. See below the added wording in the "Introduction".

Changes	In line 73: "... against monthly-aggregated country wind power generation data from the European Network of Transmission System Operators for Electricity (ENTSO-E) [24] as well as ..."																																																				
23	Section 1.10: The last part describes the effect that larger turbines were better described than low turbines: how did you determine that these were the result of power curve representation and terrain effects. I can understand why the production close to the ground is harder to predict, because the models do not resolve small scale features at these heights, but why would the power curve be less accurate for smaller turbines?																																																				
Response	Thank you. This is indeed expressed in a misleading manner. The first point refers to the synthetic power curve which is, since it is based on 106 modern turbines with a capacity of at least 1 MW [Ryberg et al. 2019 (https://doi.org/10.1016/j.energy.2019.06.052)], not well suited for small turbines. We added the following clarification:																																																				
Changes	In contrast, simulations of wind turbines with a capacity below 1 MW tend to exhibit suboptimal performance, primarily due to observed discrepancies in synthetic power curve representation - since the synthetic power curve is based on 106 modern wind turbines with a capacity of at least 1 MW - and the inherent difficulties in reducing wind speeds to a scale commensurate with the terrain.																																																				
24	Section 1.13: 22 locations: this is very little. Could you indicate if these results are statistically significant? The text should point to the data that are included in this comparison, otherwise it should not be included.																																																				
Response	Thank you. Unfortunately, due to the low number of locations the results are not statistically significant. We would have preferred to simulate more placements. This was not possible due to the limitations of the other tools which do not cover most of the locations either geographically or with regard to the turbine model. However, we think that the comparison still provides interesting insights, which is why we included it in the appendix. We added a table in the Supplementary which lists the wind farms and their respective coordinates that were used for the comparison.																																																				
Changes	Table 1: Indicative locations and capacities of the wind farms used for comparison    Wind farm No. Latitude [°] Longitude [°] Turbine capacity [kW]    164.901510.88882300 263.50128.76824200 357.25909.66373300 457.24799.68263300 555.145811.96203300 655.148011.95863300 759.31574.9048600 854.808211.50003300 958.15576.68182300 1056.49709.19263000 1156.50389.20133000 1264.224810.37462300  	Wind farm No.	Latitude [°]	Longitude [°]	Turbine capacity [kW]	1	64.9015	10.8888	2300	2	63.5012	8.7682	4200	3	57.2590	9.6637	3300	4	57.2479	9.6826	3300	5	55.1458	11.9620	3300	6	55.1480	11.9586	3300	7	59.3157	4.9048	600	8	54.8082	11.5000	3300	9	58.1557	6.6818	2300	10	56.4970	9.1926	3000	11	56.5038	9.2013	3000	12	64.2248	10.3746	2300
Wind farm No.	Latitude [°]	Longitude [°]	Turbine capacity [kW]																																																		
1	64.9015	10.8888	2300																																																		
2	63.5012	8.7682	4200																																																		
3	57.2590	9.6637	3300																																																		
4	57.2479	9.6826	3300																																																		
5	55.1458	11.9620	3300																																																		
6	55.1480	11.9586	3300																																																		
7	59.3157	4.9048	600																																																		
8	54.8082	11.5000	3300																																																		
9	58.1557	6.6818	2300																																																		
10	56.4970	9.1926	3000																																																		
11	56.5038	9.2013	3000																																																		
12	64.2248	10.3746	2300																																																		

	13	54.8384	11.3002	3000
	14	56.5004	9.1969	3000
	15	58.7304	5.9481	3450
	16	56.7656	8.6697	3000
	17	63.8198	9.6300	2300
	18	64.2155	10.4167	3300
	19	54.8137	11.4983	3300
	20	54.8318	11.3026	3000
	21	55.1435	11.9654	3300
	22	56.7567	8.6451	3000

25 Section 1.13: Fig 10 and Table 2: why is there 22 data points in Fig 10 and 21 in Table 2?

Response Thank you. We have corrected the typo in Table 2. .

Changes

Table 2:

Indicator [unitless]	EMHIRES	ETHOS.RESKit	RenewableNinjas
Root-mean square error	0.227	0.149	0.165
Pearson correlation	0.768	0.897	0.878
Detrended cross-correlation analysis (DCCA) coefficient	0.654	0.847	0.819
Perkins skill score	0.8663	0.8754	0.7810
Mean error	-0.0020	-0.0123	0.0031
number of locations	22		
total amount observations [years]	98		

26 Section 1.13: Where is this comparison performed on the github examples you provided?

Response Thank you. It is indeed not contained in the *ETHOS.RESKit* github repository since we are not legally able to openly share the proprietary location and turbine design data for the 22 locations provided by theWindPower.net. So, the code would not be executable to the open-data user. The data is commercially available to everyone though. Once one has purchased the dataset, this example script on github can be used to reproduce our results: <https://github.com/FZJ-IEK3-VSA/RESKit/blob/master/examples/ETHOS.RESKit.Wind.ipynb>
The extraction of the renewables.ninja data for comparison is done manually from the website (<https://www.renewables.ninja/>), and EMHIRES

	(European Meteorological derived High Resolution RES generation time series for present and future scenarios) is an Excel dataset that can be downloaded from their page: https://data.jrc.ec.europa.eu/dataset/jrc-emhires-wind-generation-time-series . In order to further ease the reproduction of our results once one has the data from theWindPower.net, however, we have now added a table in the appendix section 1.13 with the indicative locations and capacities of every of the 22 wind farms used for the comparison.																																																																																												
Changes	Locational and turbine model information about the particular wind farms used for the above comparison are part of the proprietary dataset that can be purchased from thewindpower.net, so the exact locations and designs cannot be shared here. To enable reproducibility of the results for users with access to thewindpower.net data, the following Table 3 contains the indicative locations of the wind farms as well as the capacity. Table 2: Indicative (rounded) locations and capacities of the wind farms used for comparison    Wind farm No. Latitude [°] Longitude [°] Turbine capacity [kW]    164.901510.88882300 263.50128.76824200 357.25909.66373300 457.24799.68263300 555.145811.96203300 655.148011.95863300 759.31574.9048600 854.808211.50003300 958.15576.68182300 1056.49709.19263000 1156.50389.20133000 1264.224810.37462300 1354.838411.30023000 1456.50049.19693000 1558.73045.94813450 1656.76568.66973000 1763.81989.63002300 1864.215510.41673300 1954.813711.49833300 2054.831811.30263000 2155.143511.96543300 2256.75678.64513000  	Wind farm No.	Latitude [°]	Longitude [°]	Turbine capacity [kW]	1	64.9015	10.8888	2300	2	63.5012	8.7682	4200	3	57.2590	9.6637	3300	4	57.2479	9.6826	3300	5	55.1458	11.9620	3300	6	55.1480	11.9586	3300	7	59.3157	4.9048	600	8	54.8082	11.5000	3300	9	58.1557	6.6818	2300	10	56.4970	9.1926	3000	11	56.5038	9.2013	3000	12	64.2248	10.3746	2300	13	54.8384	11.3002	3000	14	56.5004	9.1969	3000	15	58.7304	5.9481	3450	16	56.7656	8.6697	3000	17	63.8198	9.6300	2300	18	64.2155	10.4167	3300	19	54.8137	11.4983	3300	20	54.8318	11.3026	3000	21	55.1435	11.9654	3300	22	56.7567	8.6451	3000
Wind farm No.	Latitude [°]	Longitude [°]	Turbine capacity [kW]																																																																																										
1	64.9015	10.8888	2300																																																																																										
2	63.5012	8.7682	4200																																																																																										
3	57.2590	9.6637	3300																																																																																										
4	57.2479	9.6826	3300																																																																																										
5	55.1458	11.9620	3300																																																																																										
6	55.1480	11.9586	3300																																																																																										
7	59.3157	4.9048	600																																																																																										
8	54.8082	11.5000	3300																																																																																										
9	58.1557	6.6818	2300																																																																																										
10	56.4970	9.1926	3000																																																																																										
11	56.5038	9.2013	3000																																																																																										
12	64.2248	10.3746	2300																																																																																										
13	54.8384	11.3002	3000																																																																																										
14	56.5004	9.1969	3000																																																																																										
15	58.7304	5.9481	3450																																																																																										
16	56.7656	8.6697	3000																																																																																										
17	63.8198	9.6300	2300																																																																																										
18	64.2155	10.4167	3300																																																																																										
19	54.8137	11.4983	3300																																																																																										
20	54.8318	11.3026	3000																																																																																										
21	55.1435	11.9654	3300																																																																																										
22	56.7567	8.6451	3000																																																																																										

Reviewer #2

1	What are the noteworthy results?  • The introduction of the validated tool ETHOS.RESKitWind for global wind power generation simulation, including a library with 880 turbine types as well as support for the creation of customisable synthetic power curves. • The innovative calibration process, which uses over 8 million wind speed measurements from global meteorological mast sites across 25 countries and more than 8 million hours of power generation data from 152 wind turbines across seven onshore and offshore regions.
---	---

	Will the work be of significance to the field and related fields? How does it compare to the established literature? If the work is not original, please provide relevant references.  • The developed tool improves on existing global wind energy simulation tools by including regional calibration factors. Does the work support the conclusions and claims, or is additional evidence needed?  • Yes, the claims are backed up with the results. Are there any flaws in the data analysis, interpretation and conclusions? Do these prohibit publication or require revision?  • The difference between wind power and wind energy should be discussed, as I feel it is not used consistently. You talk a lot about power, but capacity factor is a reflection of the energy production.
Response	Thank you. In the context of wind turbine simulations, it is indeed important to distinguish between wind power and wind energy. Wind power refers to the instantaneous output of a turbine and is influenced by factors such as wind speed and turbine characteristics at a given moment. In contrast, wind energy refers to the accumulated electrical output over a period of time. The terms were indeed not used consistently. Since we focus on simulating time-resolved power output of turbines, we try to use the phrase wind power wherever possible. Wind power generation is also a commonly used term across literature. We therefore carefully checked the paper and changed the following occurrences:
Changes	In line 40: Thus, evaluating extractable wind energy resources is essential to develop strategies for the energy systems transformation, [...]. In line 45: Extractable wind energy resources depend on the location (spatial dependency), on the conditions at a particular time (temporal dependency), [...]. In line 311: Additionally, the inclusion of regional correction factors enhances the precision of wind energy assessments, even in areas currently lacking wind turbine installations Table 1: Comparison of common global open-source wind power models. In line 291: These advancements position ETHOS.RESKit as a leading open-source tool for global wind power modeling. In line 297: Future work should concentrate on the resolution of these remaining biases in order to further enhance the precision of wind power simulations. In line 307: ETHOS.RESKit marks a major step forward in wind power modeling, combining global applicability with high spatial resolution and the capability to simulate a wide range of technical turbine characteristics. As the first wind power simulation tool to undergo a rigorous validation and

	calibration process across diverse spatial and temporal scales on a global level, it sets a new standard in the field. Additionally, the inclusion of regional correction factors enhances the precision of wind energy assessments, even in areas currently lacking wind turbine installations. In line 318: In this section, we outline the comprehensive methodology employed for our wind power simulation and validation approach implemented in ETHOS.RESKit [10,26] (see more details about ETHOS.RESKit in Supplementary material 1.6), aimed at providing robust basis for global wind energy assessments. In line 321: The methodology is structured into four subsections covering (a) data acquisition and processing, (b) deriving global wind speed calibration factors aiming at addressing potential mean errors in the underlying weather data, (c) a subsequent extensive validation of our wind power simulation workflow by comparing against time-resolved park level power generation data, country-level power generation data and national statistical data, and (d) deriving national correction factors. In line 322: In the following, we will address the acquisition, classification, and processing of data crucial for the validation and enhancement of our wind power simulation workflow.
2	In this paper, the developed tool is stated to be transparent, open source, validated and evaluated. The relevance and need for an “open source” tool is not discussed. Please explain this in the paper.
Response	Thank you. We have highlighted the relevance of open-source models and the reasoning in favor of such models in the text. Please see below additions.
Changes	In line 53: A methodological gap persists, however, when it comes to the calibration and validation of such wind power and energy simulation models. For example, a previous study [8] found that only 21% of studies assessing large-scale wind resource potentials conduct a validation of the input data. For the open-source simulation tools, the situation is even worse: Whilst renewables.ninja provides a calibration over selected European countries [9], no wind energy simulation tool is validated at global scale to the knowledge of the authors. The consequence is a lack in reliability of the simulation results. The present paper addresses this shortcoming by developing a global calibration and validation process which is demonstrated using the example of the open-source ETHOS.RESKit but is applicable also to other wind energy simulation tools. The importance of open-source models and datasets in energy research has gained increased attention, as they facilitate transparency, enable reproducibility of results, and promote collaborative development [8].

3

Is the methodology sound? Does the work meet the expected standards in your field?

The methodology is generally sound and meets the expected standards in the field.

Please include a discussion about if a national split is really the best way to split the data. Did you look at other factors, such as terrain type?

Response

Thank you. We agree with your recommendation and have tried to do that. The major challenge here for us was, however, that turbine output data is available either at high-resolution but scarcely, or globally but only in the form of national annual capacity and energy output, i.e. average capacity factors. The high-resolution turbine data was by far insufficient to reliably derive statistical trends.

We did, however, assess the impact of terrain and land cover on the global bias of wind speeds at 10m resolution (measured vs. GWA). This analysis was not included in the paper so far, but we have now added a figure with discussion in the Supplementary section 1.12 based on your recommendation.

Changes

In the Supplementary Material:

Figure 1. Deviation between the average wind speed from global 10m station measurements from HadISD dataset and corresponding GWA3 values for different land types.

The comparison also revealed error trends in mean wind speeds in GWA3 10m data depending on the ruggedness of the terrain as well as on the land cover type and latitude (see Figure 11 and Figure 10): Figure 10 reveals a similar pattern as Figure 9. Especially near the equator the mean measured wind speed and the mean wind speed reported by the GWA differ by more than 67%. Generally, the mean wind speeds as reported by the GWA tend to

	be lower than the measured values. Differences in accuracy of the GWA depending on the land type can also be observed. While the GWA underestimates wind speeds in e.g. forest and urban terrain, wind speeds in areas with snow are greatly overestimated.
4	Please explain how you accounted for electrical losses, which are present in measurement data but not in the simulations? I assume these are part of the calibration, but maybe there is some variation over wind speed, turbine type, location, wind farm size, etc.
Response	Thank you. Within the workflow we account for wind-speed dependent wake losses and downtimes. Downtimes are accounted for by an availability factor. We do not explicitly account for turbine or site-specific electrical losses, since we do not have sufficient data for that. This is now mentioned in the Method section “Comparison with time-resolved wind turbine power generation data”. Other losses such as electrical losses are indirectly included within the capacity factor correction raster (which is based on comparison against official IEA data) that we provide alongside the paper. This capacity factor correction raster is described in the Method section: “Comparison with country-level statistical data”.
Changes	In line 483: Maintenance-related and other losses (e.g. electrical) are highly location dependent and not available. Therefore, other losses are not included. In the preprocessing step maintenance times were already filtered out.
5	Please include a description of which database technology you used, how it was implemented, and how the data was structured and managed.
Response	Thank you. Generally, we harmonized and processed all of our time-resolved input data. After processing the data using the xarray python package, we saved data into one netCDF file per location (and measurement height). We added the following parts to the methodology:
Changes	In line 344: Measurements were harmonized following the steps outlined in section 1.2 of the supplementary material. In line 346: For further processing, we resampled the measured wind speeds and those from ERA5 to hourly values, standardized them to UTC time, and saved their geolocations as well as the measurement heights respectively within one netCDF file per measurement height. In line 348: Lastly, the processed data was saved in one netCDF file per location. In the appendix S1.2: As the data formatting, level of detail and temporal resolution varies among

	the data sources, the meteorological data was standardized (UTC time convention, averaged one-hour resolution) using the xarray python package [1]. [...] Afterwards, the measurements were saved into one netCDF file per location and measurement height. In the appendix S1.3: The formatted data was also combined into one data set using the xarray python package [1]. [...] Finally, the processed data was saved into one netCDF file per location.
6	On line 477, you refer to “real power curves”. Can you please clarify if this means “real theoretical power curves” or “real measured power curves”? I suspect it’s the first option, in which case you should comment on the fact that the actual site-specific power curves may vary from the theoretical power curves. Also, please explain how turbine-specific synthetic power curves were generated.
Response	Thank you. You are right, we are indeed talking of real theoretical power curves, which can deviate from the measured ones depending on age, manufacturing quality, installation, maintenance etc. We have clarified that with an additional sentence now. Also the generation of the synthetic power curves is explained briefly now. For the generation of the synthetic power curves we employ the methodology outlined in [Ryberg et al. 2019 (https://doi.org/10.1016/j.energy.2019.06.052)]. Here, 130 manufacturer power curves were used to create a fit between wind speed and capacity factor based on specific power. By supplying any specific power a synthetic power curve is generated from this fit.
Changes	In line 475: First, ETHOS.RESKit was utilized to simulate the wind turbine power generation time-series for the measured time spans of each real turbine considering their specific hub heights, rotor diameters, and real power curves, if available. Those power curves are based on the information provided by the manufacturers and may be considered theoretical compared to the measured power curve of an installed turbine on site. In cases where real manufacturer power curves were unavailable, a synthetic power curve was generated based on the specific power and a fit derived from 130 manufacturer power curves [10].
7	Is there enough detail provided in the methods for the work to be reproduced?  • Yes, and the Python toolkit is available on GitHub. I haven’t tried to use it, though.

Response	Thank you.
Change	-
8	Comments on the abstract: The abstract needs to be more specific: It's not clear what you mean exactly with "Wind power simulations to support deployment strategies vary drastically in their results, hindering reliable design decisions." Which "wind power simulations" are you referring to? And with "design", are you referring to wind turbine design or wind farm design or both? These are two very different things. Please rephrase to take account of this.
Response	Thank you. We have clarified the sentence. Indeed we are more focused on wind farm siting. We have adapted our wording to convey our statement more clearly here.
Changes	In line 19: Wind power is expected to play a crucial role in future net-zero energy systems, but existing wind resource and power simulations to support deployment strategies of wind farms vary drastically in their results, hindering reliable wind farm siting and system integration design decisions.
9	It's not clear what the need for a transparent, open source, validated and evaluated, global wind power simulation tool is. Please explain this.
Response	Thank you. We have now addressed this point in the introduction.
Changes	In line 53: A methodological gap persists, however, when it comes to the calibration and validation of such wind power and energy simulation models. For example, a previous study [8] found that only 21% of studies assessing large-scale wind resource potentials conduct a validation of the input data. For the open-source simulation tools, the situation is even worse: Whilst Renewables.ninja provides a calibration over selected European countries [9], no wind energy simulation tool is validated at global scale to the knowledge of the authors. The consequence is a lack in reliability of the simulation results. The present paper addresses this shortcoming by developing a global calibration and validation process which is demonstrated using the example of the open-source ETHOS.RESKit but is applicable also to other wind energy simulation tools.

	In line 68: Validation, as understood by the authors, involves comparing model outcomes with real-world data to assess how accurately the simulation represents actual observed conditions.
10	Please clarify what the “global wind power simulation tool” actually simulates.
Response	Thank you. We have extended the following sentence to make clear that our global wind power simulation tool simulates the time-resolved energy output of any given wind turbine at a user-defined location.
Changes	In line 21: Therefore, we present a transparent, open source, validated and evaluated, global wind power simulation workflow for the renewable energy simulation tool ETHOS.RESKit . It enables time-resolved simulations of wind turbine energy output with high spatial resolution and supports customizable designs for both onshore and offshore wind turbines.
11	Please quantify how high the spatial resolution is. And are the 22 customizable designs for wind turbines or for wind farms?
Response	Thank you. The spatial resolution of the hourly patterns alone is 0.25° x 0.25° as per ERA5 resolution. The wind speed scaling based on the high-resolution Global Wind Atlas data divided by the resampled ERA5 long-run averages (please also refer to the comment 15 of reviewer 1), however, creates an effective resolution of 250m x 250m by downsampling, with individual wind speed curves per each 250m cell. We have also clarified this in the text now. The number 22 occurs only once in our assessment when comparing our model to other available models. Here, 22 occurs as the number of wind farms (not wind turbines) which are covered by all 3 models in the comparison (geographically and turbine model-wise). – The number of “designs for wind turbines” in RESKit is much greater or even unlimited, however, with 123 turbine models in the open-data free version on github alone, 880 real turbine models in the (proprietary) database of thewindpower.net, and an unlimited number of custom turbine designs via the free choice of turbine capacity, hub height and rotor diameter in combination with the synthetic power curve generator.
Changes	In line 270: ETHOS.RESKit leverages high-resolution wind data (250 m x 250 m) from ERA5 and GWA3
12	Please clarify the statement “We achieve a global average capacity factor mean error of 0.006 and Pearson correlation of 0.865.”. Does this refer to a comparison to energy yield measurements? Over what time periods? Where were all the projects? This is difficult to understand, and the abstract should be understandable independently of the rest of the paper.

Response	Thank you. The global average capacity factor mean error and Pearson correlation refer indeed to the comparison to energy yield measurements. The time period is wind farm specific. In the Method section, we specify that the measurements range from 2002 to 2021 from 6 countries in total. Figure 7 also highlights the locations of the wind farms in a map. We have added the following to add clarity.
Changes	In line 23: The tool has undergone an extensive validation and calibration process using over 16 million global measurements from meteorological masts for wind speed bias correction and 8 million measurements from wind turbine sites across 6 countries during 2002 to 2021. In comparison with measured energy yield from wind turbine sites, we achieve a global average capacity factor mean error of 0.006 and Pearson correlation of 0.865.

Reviewer #3

1	The paper presents a transparent, open source, validated and evaluated, global wind power simulation tool called ETHOS.RESKitWind with high spatial resolution and customizable designs for both onshore and offshore wind turbines. The following comments are for the authors of the paper. The paper did not specify the types of wind turbines that the authors claimed they simulated in the abstract and introduction. According to the authors 800 wind turbines were simulated, and which type of analysis did they carry out during the simulation?
Response	Thank you. This is a potential misunderstanding. The over 800 turbines that are mentioned in the Introduction are the full range of real turbine model parameter sets that are available in ETHOS.RESKit for simulation to the user, in addition to the synthetic power curves. In the present assessment, we have specifically simulated the real existing model fleet in the world, as exactly as possible. This employed over 490,000 turbines with 439 different models in total, determined by the type of models that are still in use globally and the quality of information in the used wind farm information dataset (sometimes, “only” diameter, hub height and capacity were available, and we have used synthetic power curves for this model as described). The hourly output time-series for every year of every one of those turbines were then simulated, and later an analysis of the hourly correlation as well as an analysis of the annual energy balances was done, compared to real

	generation data, to assess the quality of temporal patterns and total energy generation. We have updated our text now to reflect these differences better.
Changes	In line 115: Our model addresses the identified limitations through extensive validation, global applicability, and the incorporation of more than 800 wind turbine models available to the modeler. To enhance precision, we implement a comprehensive calibration of wind speed data gathered from 213 global weather mast locations in 25 different countries globally, spanning over 8 million hours of observation after filtration. Furthermore, we validate the simulated wind power output by comparing it with the actual hourly output from 152 turbines and wind farm sites. Finally, we further validate our model by comparing the outcomes of the simulation of over 30 000 existing global wind farms with over 490 000 turbines with publicly available country-level hourly wind power generation data, as ...
2	The paper is not written as a standard Journal paper. From Introduction then Results, no models, no theory, no descriptions of control to help bring out the methodology and technical aspects of the paper. The paper is scattered and not well structured. Method employed is coming after Results.
Respo	Thank you. For this, we have to adhere to Nature's guidelines and therefore do not have much flexibility.
Changes	Please refer to the changes made in response to comments 1, 2, 4 and 13 by Reviewer 1 and comments 4 and 6 by Reviewer 2, where we have added additional information and clarification regarding the motivation and methodology.
3	Besides, there is no typical model of the wind farm with detailed parameters and ratings for easy replication of the study. The presented results are limited and not showing the performance of some of the key variables of the wind turbines employed.
Response	Thank you. We regret very much that the input data for global turbine locations and design parameters is proprietary and we can therefore indeed not openly share the input data, but there is simply no open dataset available with comparable accuracy and completeness. However, we offer an example script on github which reproduces our results if the processed database from theWindPower.net (which is commercially available to everyone) is used as input. In a simplified example case, it demonstrates both the use of real wind turbine model power curves and synthetic power curves, generated based on the specific turbine capacity: https://github.com/FZJ-IEK3-VSA/RESKit/blob/master/examples/ETHOS.RESKit.Wind.ipynb. We have, however, clarified our legal constraints in a remark in the paper to explain

why we are not sharing this and where the data is available (commercially to everyone).

However, some of the wind turbine data with hourly measurements are openly available. Supplementary Table 4 lists all data sources used in this study. This table also lists hub-height, capacity and rotor diameter ranges for each of the used input data sources.

In table 2 of the paper, we present key statistical indicators that include the measured and simulated mean capacity factor of the wind turbines.

Furthermore, we evaluate and show the mean error, Perkins skill score, RMSE, Pearson correlation against the measured and simulated capacity factors.

For the comparison against other tools in the Supplementary material we added the locations and capacities of the used wind farms.

Changes

In line 369: Additionally, we acquired a **proprietary** database on existing wind farms containing data on 26,900 wind farm locations worldwide as well as databases on turbine models and power-curves **available from thewindpower.net** [25] to simulate the existing wind fleet stock and derive national correction factors.

In the Supplementary material:

Locational and turbine model information about the particular wind farms used for the above comparison are part of the proprietary dataset that can be purchased from thewindpower.net, so the exact locations and designs cannot be shared here. To enable reproducibility of the results for users with access to thewindpower.net data, the following Table 3 contains the indicative locations of the wind farms as well as the capacity.

Table 3: Indicative locations and capacities of the wind farms used for comparison

Wind farm No.	Latitude [°]	Longitude [°]	Turbine capacity [kW]
1	64.9015	10.8888	2300
2	63.5012	8.7682	4200
3	57.2590	9.6637	3300
4	57.2479	9.6826	3300
5	55.1458	11.9620	3300
6	55.1480	11.9586	3300
7	59.3157	4.9048	600
8	54.8082	11.5000	3300
9	58.1557	6.6818	2300
10	56.4970	9.1926	3000
11	56.5038	9.2013	3000
12	64.2248	10.3746	2300
13	54.8384	11.3002	3000
14	56.5004	9.1969	3000
15	58.7304	5.9481	3450
16	56.7656	8.6697	3000
17	63.8198	9.6300	2300
18	64.2155	10.4167	3300

	19	54.8137	11.4983	3300
	20	54.8318	11.3026	3000
	21	55.1435	11.9654	3300
	22	56.7567	8.6451	3000
S	The code looks fine.			
R	Thank you.			
Cha	-			

Dear reviewers of the manuscript “Towards high resolution, validated and open global wind power assessments” with ID: NCOMMS-24-84206,

We again thank you for your time and your constructive comments to improve the manuscript. We confirm that we have addressed each comment in the revised manuscript as outlined below. Please find additional minor adaptations for improved readability and better understanding of the considered losses highlighted as tracked changes in our manuscript and supplementary material.

We have structured our answers in a tabular format such that first your questions are listed in blue, in the green box below we answer your question and summarize the resulting changes from the manuscript and supplementary material finally in the red box below. Any reference to line numbers in the replies below refers to the originally submitted version from the first review to avoid confusion.

Reviewer #1 (Remarks to the Author):

1	General comments The paper is relevant and outlines a method to obtain capacity factors for energy planning purposes. There is a lot of information in the paper, although the structure is partially still confusing due to the nature format with supplementary materials. The code is nicely documented and the authors have done a great job adding a toolchain to achieve many things.
Responses	Thank you again for your input and the kind words.
Changes	
2	The global wind atlas 4 has been released (https://doi.org/10.11583/DTU.28955267), so in a way the paper contains a outdated dataset. Ideally it would be nice to redo the results using this improved version but can understand if that is a lot of work. So if this is not possible, it would be good to comment on this in the introduction or discussion. In any case I would probably notify the reader which Global Wind Atlas version is used by writing for example GWA3 instead of GWA.

Response

Thank you, also for your kind understanding of the implied effort. To be honest, we have struggled with the decision given the significant amount of effort and calculation time but have finally decided to recalculate the whole workflow with GWA4. We share your view of the significance of GWA4 and wish to maximize the relevance of our publication by including this latest development. GWA4 also indeed showed some interesting improvements over GWA3 in our comparisons! We found that using GWA4 reduced the mean wind speed at our weather masts from 7.21 m/s to 7.11 m/s (for a measured mean of 7.07 m/s, GWA shear projection applied in both cases), reducing the MAE from 1.76 to 1.72 m/s. The overall qualitative trends were the same, however, which means that all messages of the paper remain true, with only slightly adapted quantitative results. Note that only the best fitting wake reduction curve changed from “dena mean” to “knorr mean”, as visualized in SM Fig. 7. As a general conclusion, we can say that the GWA4-corrected version tendentially yields higher values than the GWA3-corrected version, which is equalized by slightly different cf-correction factors in the last step then in order to achieve realistic national average cf. This means that some key indices of location-sharp comparisons may have looked even (slightly) better with the earlier GWA3 version. This is an expected outcome, however, given that the new workflow has less bias towards Northern latitudes (and particularly Europe): The earlier version had less mean error e.g. in Europe, but was even further off in the Global South. The latter has been improved a lot, but comes at the cost of a slight overestimation in Northern latitudes due to the general GWA4 latitude pattern. Locations with individual hourly or annual data are mainly in such regions though, leading to the observed overestimations in location-sharp capacity factor data. As said, the ultimate impact is marginal though, given that the final workflow step allows to relatively correct avg. cfs to the expected statistics.

Several sections of the paper have therefore been updated with the adapted results, but remained consistent in their core statements, see below:

Changes

(multiple changes from GWA3 -> GWA4 throughout the paper and supplementary)

(Update of Figure 1, Table 2, Figure 2, Figure 3 and Figure 5 in the manuscript)

(Update of Figure 1, Table 1, Figure 8, Figure 11 and Figure 12. Addition of Figure 2 and Figure 7 in the supplementary).

(Values within the text have also been updated accordingly)

In line 112ff: In addition, global mean errors in wind speeds at 10 m were also found in the GWA4 (see Supplementary 1.12). An overall underestimation across the globe is even more pronounced in Eastern Australia, along the US coastlines, in Southern Europe and the Mediterranean, the Siberian Taiga and rain-forested regions, to a lesser extent also throughout Sub-Saharan Africa and South America. Overestimations occur isolated especially in the Northern hemisphere everywhere, but are found in larger quantities throughout continental Asia and in Northern Europe as well as in Canada (see Supplementary Figure 2).

In line 164ff: Figure 1 (a) shows the result of a calibration applied to simulated wind speeds. The calibration increases wind speeds below approximately 4.0 m/s and decreases those above it. Figure 1 (b) details the non-linear nature of this adjustment, showing the correction factor peaking at approximately 12.5 m/s. At this peak, the original wind speed is adjusted downwards by about 11%. As speeds increase beyond 12.5 m/s, the magnitude of the correction gradually lessens until 17.5 m/s.

In line 220ff: Wind parks in Denmark, Belgium and the USA show both negative mean errors (underestimation) and positive mean errors (overestimation). Wind parks the Netherlands, on the other hand, predominantly exhibit underestimation.

In line 227ff: After minimizing the effects of technology differences, temporal uncertainties, and locational variations, the model showed a very good match on global average, with national discrepancies of mostly below 10%. The IEA reports a global average capacity factor of 0.306 across 71 countries and offshore regions, while the model yielded an average of 0.287, a relative deviation of 6.2%. In comparison, the non-calibrated workflow demonstrated a significant overestimation, with an average capacity factor of 0.377 and a relative deviation of 23.1%.

Regional trends are depicted in Figure 5. At global scale, the pattern largely follows the GWA4 trend of underestimation towards and just South of the equator (cf. Supplementary 1.12). This also means that global correction cannot be perfect with a single set of windspeed correction factors. The small

tendency for overestimation in Northern latitudes hence aligns with the findings of the location-specific comparisons at windfarm level (see above). Whilst the errors here are reduced significantly by the wind speed correction, the multinational distribution of the weather masts for calibration does perfectly reflect the specific Northern latitude trends. This in turn leads to slightly overestimated capacity factors in Northern and extreme Southern latitudes, with locally more pronounced patterns due to various reasons. Lower underestimation by GWA4 in Northwestern Europe might explain deviations in Great Britain and Germany, whilst individual outliers such as Panama, the Japanese offshore locations or Cyprus may be caused by external factors: The IEA dataset provides national energy and capacity values per year only but lacks information about individual windfarms and technology characteristics, necessitating assumptions and external data sources to define turbine properties. Further uncertainties stem from both the national and the annual averaging of generation data, which obscures spatial and temporal dynamics, and from challenges in precisely locating turbines or identifying their commissioning dates or missing design parameters. Additionally, external influences such as grid congestion, curtailment, import/export dynamics, and discrepancies in reporting conditions contribute to differences between simulated and actual results.

Figure 5: Capacity factor deviation map between ETHOS.RESKit and IEA data [24] based on the average deviation in the years 2017 to 2021.

- 3 The paper is improved, but there is still methodological problems. In step 4 of the supplementary materials (l226) you are using the local roughness length to extrapolate around hub height. This is not valid in the majority of onshore cases. At 10 m height, using the local roughness length may work quite well, but this becomes very problematic at 100 m, because the local roughness length is often irrelevant, because the footprint at 100 m is typically several kilometers upstream of the current wind direction. For example, we are near the coast and the local roughness is 1 m (forest), but the wind is coming 95%

	from offshore directions. In that case the roughness length should be close to 0.0002 m and not 1 m! The proper way to do this is to use the mesoscale roughness length and speedup-factors due to internal boundary layers, but this may be beyond the scope of the paper. An alternative is to revise this extrapolation method and use the observed wind shear from the different heights of the global wind atlas. But it is well documented that vertical extrapolation does not follow the log-law at greater heights (e.g. https://link.springer.com/article/10.1007/s10546-007-9166-9).
Response	Thank you. We agree that the wind speeds at turbine height are influenced by wider areas beyond the immediate local roughness factor. We have therefore tested your suggestion and found that using a linear interpolation between the GWA height levels indeed improves the accuracy of the wind simulations, in the case of our mast collection, the mean ws was changed from 7.35 m/s down to 7.21 m/s (for a measured mean of 7.07 m/s, here still for GWA3) with MAE reduced from 1.82 m/s to 1.76 m/s. This has further influenced our decision in favor of recalculating the whole workflow, in addition to the GWA4 question. Reason is that a change of the height scaling approach required complete recalculation since it affects nearly every step, starting as early as scaling the wind speeds to the mast heights for the initial wind speed calibration. We have also implemented the option to extrapolate to hub height using the observed wind shear from different heights of the global wind atlas into the workflow of ETHOS.RESKit which is available on GitHub. We added the following parts to the respective sections of the paper. We would like to explicitly express our gratitude for this meaningful contribution to the quality of our work here and have also added a statement to the acknowledgements.
Changes	In the acknowledgements: We are grateful to the unknown reviewers who have contributed to the quality of this analysis with their comments and questions. Our special thanks go to Reviewer #1 who suggested the wind speed shear approach based on the GWA4 that we ended up using over the logarithmic scaling based on surface roughness. In the supplementary: The corrected wind speeds are extrapolated to the hub height (or anemometer height) using the observed shear from different height levels of the GWA4. For this, a scaling factor is derived by calculating the wind speed at hub height from the GWA4 by interpolation divided by the wind speed from GWA4 at 100m.

	Simulated wind speeds are then extrapolated by multiplication with this scaling factor In the github repository: (The option to use shear/linear interpolation of long-run average data at different heights to scale windspeeds to hub heights has been added as a feature to the RESKit repository on github, and the shear of GWA4 has been set as default in the new wind simulation workflow that is introduced in the present paper)
4	I am still confused by the calibration process. Is the correction factor defined by country? Or is there one for the whole world? If the latter, the correction factors are entirely defined by the countries which are presented in the tall tower dataset. What happens if you leave out countries like Iran, for which there is many towers used in calibration. Wouldn't this totally change the correction factors? In other words: without more justification for this wind speed dependent tuning it is hard to defend that you can apply them anywhere in the world. They are fully determined by the masts used in the calibration and thus not general.
Response	Thank you. The correction factor is one for the whole world. However, different corrections are applied depending on the hourly wind speed. Please note that we tried to use all available public data of wind speed measurements at heights relevant for wind turbines. We would have liked to do a country or region-specific calibration to counter the trends describes in the paper, but there is simply not sufficient data available to do so. Therefore, we choose to derive a consistent correction factor set for the whole world. Note that while Iran has a lot of masts, the measurement time spans are a lot shorter than for the masts in Europe and USA. Since we also identified a weather mast count imbalance as a potential bias, we have decided against a mast location weighing in favor of an unweighted comparison. We therefore achieve a distribution which is more balanced than one might expect from the masts alone. Please see the exact numbers below: USA: 29.2% IRN: 17.5% Europe: 33.2% ZAF: 14.7% Rest: 5.4% We have clarified the reasons for this decision in the text, thank you. We also tested the validity of our correction factor set by leaving out all masts from Iran as requested to derive more regional correction factors. As shown in the

results below, the correction factor only differs marginally, indicating that the general trends are similar at global scale.

We have added further text in the manuscript that highlights the issue of data availability and how weighing by measurement data points per mast allowed us to pursue this approach nevertheless.

Changes	In a second step, wind speeds were binned in 0.1 m/s categories and a proportional regression per bin was used to fit processed and measured wind speeds. A weighing based on the number of mast-specific hours with measurement data was selected as it yielded the most even global distribution. Individual regressions per global region, landcover type and latitude or slope bin proved infeasible due to the limited amount of available measurement data at global scale. Comparative analyses showed similar trends for various world regions, however, which supports the validity of an averaged global regression function here. Alternative regressors were also tested...
5	Minor comments: l348 "the resolution of" -> "solving" l374: "comprehensive" -> better to leave these subjective words out l377: "robust" -> better to leave these subjective words out l517: found in the Supplementary -> where (Table/Fig) exactly? Suppl materials l438: "as shown in (...?)"
e	Thank you. We have addressed all of them as outlines below under "Changes".
Changes	Future work should concentrate on solving these remaining biases... In this section, we outline the methodology employed for our wind power simulation... ...aimed at providing a basis for global wind energy assessments.

	The resulting wind speed dependent scaling factors can be found in Figure 1 (b) and the supplementary files. as shown in Figure 10
	Reviewer #2
6	Thank you for responding to my comments so thoroughly - I am now happy with the changes and would like to accept the paper for publication
sp ha	Re C Thank you.
	Reviewer #3
7	Despite the revision, the authors still presented the results of the paper before the methods.
se ha	Respon C Thank you. We agree that this may sometimes be perceived as confusing, however, the Editor informed us that the order of the sections in the paper is according to the Nature Communication 's policy and should not be changed.
	-

Dear editor and reviewers of the manuscript “Towards high resolution, validated and open global wind power assessments” with ID: NCOMMS-24-84206B,

We again thank you for your time and your constructive comments to improve the manuscript. We confirm that we have addressed each comment in the revised manuscript as outlined below. Please find additional minor adaptations for improved readability and better understanding of the considered losses highlighted as tracked changes in our manuscript and supplementary material.

We have structured our answers in a tabular format such that first your questions are listed in blue, in the green box below we answer your question and summarize the resulting changes from the manuscript and supplementary material finally in the red box below. Any reference to line numbers in the replies below refers to the originally submitted version from the first review to avoid confusion.

EDITED 14th Dec. 2025:

Following the suggestion of the editor, we would like to add a special acknowledgement here to the reviewers, and most particularly to the idea by reviewer #1 to use different GWA altitude layers for wind speed height hub scaling here. We had initially planned to add this to the main acknowledgements section of the manuscript, such references should be avoided due to the editorial policy, however. We understand that this exchange will be made publicly available as well though and will therefore use this platform to express our acknowledgement and appreciation. Thank you all for your very valuable input.

We are grateful to the unknown reviewers who have contributed to the quality of this analysis with their comments and questions. Our special thanks go to Reviewer #1 who suggested the wind speed shear approach based on the GWA4 that we ended up using over the logarithmic scaling based on surface roughness.

Reviewer #1 (Remarks to the Author):

1	The manuscript is improved and the main issues have been addressed. There is still a lot of ad-hoc decisions in the model chain that I would have probably done differently, but that is unavoidable in a model chain that encompasses so many fields (wind resource modelling, wake modelling, power system). I
---	--

	would recommend the authors to look at the reference below for more inspiration.
Response	Thank you. We have added the reference to the introduction, where we position it relative to our work and highlight that, to the best of our knowledge, no wind-energy simulation tool has yet been validated at the global scale.
Changes	For the open-source simulation tools, the situation is even worse: Whilst Renewables.ninja provides a calibration over selected European countries ⁹ and Murica et al. ¹⁰ perform a validation of country-level generation time-series for European countries, no wind energy simulation tool is validated at global scale to the knowledge of the authors.
2	l36: remove "the most". I would argue there is other methods that are equally comprehensive (e.g. https://doi.org/10.1016/j.apenergy.2021.117794).
Response	Thank you. We have removed the phrase.
Changes	The release of ETHOS.RESKit is a step towards a fully open source and open data approach to accurate wind power modeling by incorporating the most comprehensive simulation advances in one model.
3	l181: skewed normal distribution -> it is for sure not normal but rather a weibull distribution because the wind speed can never be lower than 0.
Response	Thank you. That is indeed correct, we changed the sentence.
Changes	Additionally, the measured wind speeds exhibit a skewed distribution that is well-described by a Weibull distribution, with a mean wind speed of about 6 m/s, which is a relatively low average wind speed for wind energy installations.
4	1.2: Figures Figure 2, Figure 3, Figure 4, Figure 5 -> Figures 1,2,3,4 and 5
Response	Thank you. We have corrected the Figure naming. The first "Figure" was not supposed to be there.
Changes	and are shown in Supplementary Figure 2, Supplementary Figure 3, Supplementary Figure 4 and Supplementary Figure 5.
5	1.3: throttling -> curtailment
Response	Thank you. We have replaced throttling with curtailment.

Changes	Therefore, the filtering process for shutdowns or curtailment due to limitations in the power grid, maintenance, or other irregularities was more demanding.
6	1.6 bullet 1: Long-run average -> Long-term average is more frequently used bullet 3: long-term orographic -> long-term microscale (the GWA contains more than just orographic speedups, but also those due to roughness/stability). bullet 4: which levels? I would call this: Vertical extrapolation of wind speed
Respon	Thank you. We have incorporated your suggestions.
Changes	Long-term average (LRA): ... This factor is then applied to the resampled ERA5 time series data to improve the representation of long-term microscale effects. Vertical extrapolation of wind speed: The corrected wind speeds are extrapolated to the hub height (or anemometer height) using the observed shear from different height levels (10, 50, 100, 150 and 200 m) of the GWA4.